# The Phenomenon of Policy Churn

**Tom Schaul**
DeepMind
London, UK

**André Barreto**
DeepMind
London, UK

**John Quan**
DeepMind
London, UK

**Georg Ostrovski**
DeepMind
London, UK

`{tom,andrebarreto,johnquan,ostrovski}@deepmind.com`

## Abstract

We identify and study the phenomenon of *policy churn*, that is, the rapid change of the greedy policy in value-based reinforcement learning. Policy churn operates at a *surprisingly rapid* pace, changing the greedy action in a large fraction of states within a handful of learning updates (in a typical deep RL set-up such as DQN on Atari). We characterise the phenomenon empirically, verifying that it is not limited to specific algorithm or environment properties. A number of ablations help whittle down the plausible explanations on why churn occurs, the most likely one being deep learning with high-variance updates. Finally, we hypothesise that policy churn is a potentially beneficial but overlooked form of *implicit exploration*, which casts $\epsilon$-greedy exploration in a fresh light, namely that $\epsilon$-noise plays a much smaller role than expected.

## 1 The Phenomenon

Reinforcement learning (RL) involves agents that incrementally update their policy. This process is driven by the objective of maximising reward, and based on experience that the agent generates via exploration. The sequence of policies $\pi_0, \dots, \pi_k, \dots, \pi_T$ usually starts from a randomly initialised policy $\pi_0$ and aims to end at a near-optimal policy $\pi_T \approx \pi^*$. Ideally, steps in that sequence ($\pi_k \to \pi_{k+1}$) are policy improvements that increase expected reward.

This paper studies the amount of *policy change* that goes along with such a policy update process (for a definition, see Section 1.1). In particular, it makes the core observation that policy change *in practice* (as illustrated in some typical deep RL settings) is orders of magnitude larger than could have been expected, and stands in contrast to various reference algorithms (Sections 1.2 and 3.3).

> **Key observation 1:** The greedy policy changes much more rapidly than you probably think.[a]
> ___
> [a]As a coarse magnitude for the impatient reader: in a typical run of DQN on Atari, the greedy policy changes in $\approx 10\%$ of all states after *a single gradient update* (Figure 1 and Section 1.2).

We dub this phenomenon "*policy churn*" to highlight that most of this policy change may be unnecessary. We study the phenomenon in depth, determining the range of deep RL scenarios it appears in, fleshing out its properties, and in the process narrowing the space of potential causes and mechanisms involved using a set of ablations (Section 3).

Our second key message relates the phenomenon of churn to exploration, specifically in the context of $\epsilon$-greedy exploration (Section 2), with some more speculative ramifications in Section 4.

> **Key observation 2:** Policy churn is a significant driver of exploration.[a]
> ___
> [a]This holds both in the sense that reducing churn can reduce performance, and in the sense that explicitly adding noise becomes unnecessary in the presence of churn (i.e., $\epsilon = 0$ is viable).

36th Conference on Neural Information Processing Systems (NeurIPS 2022).

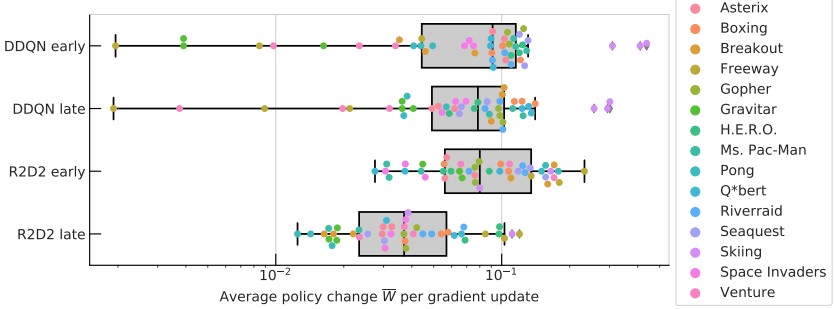

Figure 1: Average amount of policy change $\overline{W}$ (Eq. 2) per update, in two deep RL agents (DoubleDQN and R2D2). Points are averages over 3 seeds, on one of 15 (colour-coded) Atari games. "Early" denotes the first 25% of training, "late" denotes the final 25%; observe that churn magnitudes drop late in training, but only for R2D2. See Figures 15 and 16 in the appendix for more fine-grained results.

## 1.1 Defining policy change

A policy is a function from states $s \in \mathcal{S}$ to a distribution over actions $a \in \mathcal{A}$, where for the purposes of this paper $\mathcal{A}$ is discrete. We quantify the local, per-state *policy change* between policies $\pi$ and $\pi'$ using the moved probability mass (i.e., the total variation distance):

$$W(\pi, \pi'|s) := \frac{1}{2} \sum_a |\pi(a|s) - \pi'(a|s)|, \tag{1}$$

which satisfies $0 \leq W(\pi, \pi'|s) \leq 1$. When $\pi$ and $\pi'$ are *greedy* policies derived from a state-action-value function—that is, $\pi(s) \in \arg\max_a q(s, a)$—then $W(\pi, \pi'|s) = 1$ if the $\arg\max$ action in state $s$ changes upon replacing $q$ by the function $q'$ underlying $\pi'$, and $W(\pi, \pi'|s) = 0$ otherwise. Similar reasoning applies more generally when both $\pi$ and $\pi'$ are deterministic policies. We can aggregate policy change across all states, weighted by a state-distribution $d$:

$$\overline{W}(\pi, \pi') := \mathbb{E}_{s \sim d}[W(\pi, \pi'|s)], \tag{2}$$

where a reasonable choice for $d$ is the empirical state distribution encountered during training (which is non-stationary, but it could also be the stationary distribution of a fixed policy, or the uniform distribution across all states, as discussed for some of the toy scenarios below). For greedy policies (and uniform $d$), $\overline{W}(\pi, \pi')$ is simply the fraction of states where an $\arg\max$ switch occurred.

For settings where policy performance stabilises at some point $t = P$ of training (e.g., hitting a performance plateau, or converging to optimal behaviour), two additional metrics may be of interest, namely the *cumulative* policy change $\overline{W}_{0:P}$ until that point[1], and the *average* policy change after that point $\overline{W}^+$ (which could be zero, e.g., if the process converges):

$$\overline{W}_{0:P} := \sum_{i=0}^{P-1} \overline{W}(\pi_i, \pi_{i+1}) \qquad\qquad \overline{W}^+ := \limsup_{T \to \infty} \frac{1}{T - P} \overline{W}_{P:T}. \tag{3}$$

## 1.2 Quantifying the phenomenon

Given an initial and a final policy, the process with the minimum amount of policy change is an oracle that jumps from $\pi_0$ to $\pi_P$ in a single step ($P = 1$). By construction, this can incur a policy change of at most one unit, $\overline{W}_{0:P} \leq 1$. In value-based deep RL agents, a natural definition for the sequence of policies is to use the induced greedy policies $\pi_k(s) \in \arg\max_a q_{\theta_k}(s, a)$ where $\theta_k$ are the parameters of the Q-function at iteration $k$. In agents that use a target network (inducing $\pi$) that is an older copy of the online network (inducing $\pi'$), it is easy to measure $\overline{W}(\pi, \pi')$ by comparing their

---

[1]Note that the granularity of updates (e.g., batch size) determines how many intermediate policies are considered, which in turn affects the measured magnitude of policy change in learning processes that have churn.

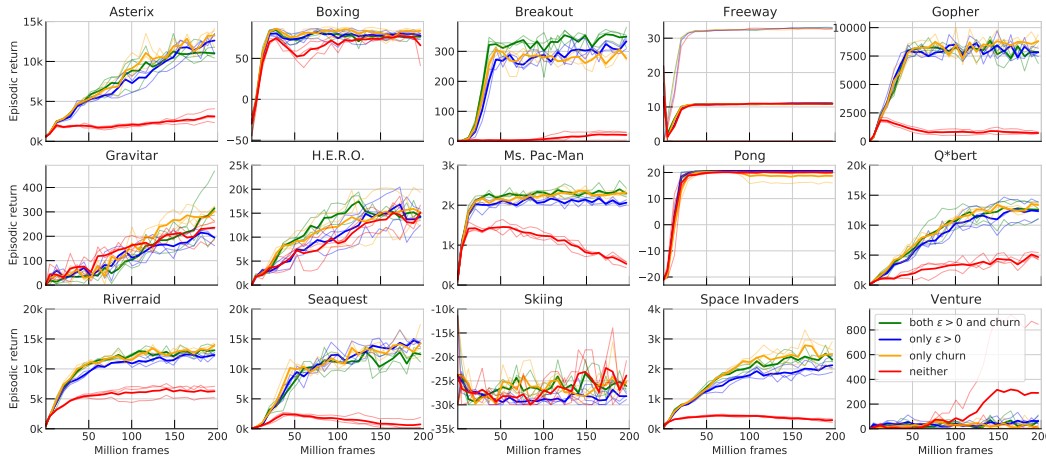

Figure 2: Impact of churn on exploration. Plots show performance of DoubleDQN on Atari, with four variants of exploration: **green** is the unmodified baseline ($\epsilon = 0.01$), **gold** changes $\epsilon = 0$, **blue** has reduced churn by acting with the target network, and **red** shows the effect of both of these changes together (no more exploration, and a corresponding performance collapse in most games). Thin lines depict individual seeds (3 per setting), thick lines are averages, smoothed over 10M frames.

$\arg \max$ actions at the points in training where the target network lags behind by just one update. Figure 1 shows typical values for $\overline{W}$ on a few Atari games, estimated by comparing the policies induced by online and (one update old) target networks, on batches of experience sampled from the agent's replay buffer. It is worth emphasising that with such rates of average policy change, of $\overline{W} \approx 10\%$ per update, the magnitude of whole-lifetime change becomes enormous: across training, an agent like DQN, which performs $10^7$ updates, changes its greedy action *a million times in each state* (on average).

A second striking result is the amount of policy change in *late* training, when the performance of the policy no longer changes (in the case of PONG it is arguably optimal): there is still a change of $\overline{W}^+ \approx 5\%$ per update (Figures 1 and 14). This highlights that a lot (maybe most) of policy churn is not directed at a policy improvement; we revisit this in Section 4.2.

**Is this unexpected?** We conducted an informal survey of over 50 deep RL practitioners, including three of the inventors of DQN [27], for their estimate on how rapidly the policy changes in a typical Q-learning based setup. The question was: `For the greedy policy to change in 10% of all states, how many learning updates does it take?` (or equivalent). The median response was $1\,000$ updates, with answers varying between 1 and $1\,000\,000$ updates. This deviates by three orders of magnitude from the empirical value of 1 update (or $\overline{W} \approx 0.1$, see Figure 1).

**Policy change in other settings.** To appreciate the large empirical magnitudes of (cumulative) policy change we observe in deep RL, we can contrast them with a few alternative settings. For example, classic dynamic programming techniques such as value iteration or policy iteration [44], when applied to toy RL domains such as FourRooms, CATCH, or DeepSea [33], accumulate $\overline{W}_{0:P} \approx 1$, not much more than the single-step oracle (see Appendix B.4). The two main differences from deep RL are their tabular nature (no function approximation, FA) and non-incremental updates. It is possible to construct tabular settings with much larger policy change, with either incremental updates (Appendix A.3) or bootstrapping (Appendix A.4). A minimalist example with non-linear FA and incremental updates is supervised learning on the MNIST dataset (the "policy" in this case are the predicted label probabilities). Training a digit classifier to convergence accumulates $\overline{W}_{0:P} \approx 10$, that is, the average input goes through 10 label switches (see Appendix B.5). None of these examples are fully satisfying, as they are not apples-to-apples comparisons; so in Section 3.3 we construct a spectrum of algorithmic variants that spans from tabular policy iteration (without churn), via tabular Q-learning to an approximation of DQN (with realistic magnitudes of churn).

## 2 The Exploration Effect

We now turn to the impact of policy churn on agent behaviour: What happens when *acting* according to policies produced by a learning process that induces rapidly changing policies? In other words, what is the effect of policy churn on exploration? While each individual greedy policy would lead to a very narrow set of experience in a (nearly) deterministic environment like Atari, the fact that the greedy policy changes so rapidly (in $10\%$ of states per update, with one update every 16 environment frames in DoubleDQN) makes for a broad data distribution. And in many circumstances this is sufficient for good performance, even in the absence of any other form of exploration, such as stochasticity introduced via an $\epsilon$-greedy policy. Figure 2 shows this across a range of Atari games: compare **green** (baseline) and **gold** ($\epsilon = 0$) curves.[2]

Conversely, removing (some of the) churn in the behaviour during training, which can be done by acting with the target network (updated only every 120 000 frames in the DoubleDQN agent) instead of the online network, sometimes reduces performance even in the presence of $\epsilon$-greedy exploration (**blue** in Figure 2).[3] Additionally, we show that performance often *collapses* completely when both forms of exploration are removed ($\epsilon = 0$ and no churn, in **red**). Figure 2 compares all four variants of exploration, indicating that the two sources of exploration have different contributions in different games.

**Sufficient exploration with $\epsilon = 0$.** The perhaps unintuitive observation of successful exploration with purely greedy policies has been made before, albeit implicitly. In particular, in the presence of certain alternative exploration methods such as noisy nets [15], no significant additional advantage is obtained from using $\epsilon > 0$ [20]. Other works containing experimental variants with $\epsilon = 0$ demonstrated successful training in this setting [35, 39], but did not highlight the result.

**Consistent behaviour and Thompson sampling.** In considering the potential exploration benefits of a rapidly changing policy, it is worth qualitatively contrasting the resulting behaviour with that of an $\epsilon$-greedy policy with $\epsilon > 0$. The latter generates high-frequency dithering, with uncorrelated random action decisions at consecutive states and the effect of most exploratory actions likely undone by the following greedy action [11]. By contrast, policy-churn-induced exploration can be expected to generate temporally correlated, consistent exploration (necessary, though perhaps not sufficient, to perform "deep" exploration, as in ensemble and Thompson sampling methods [45, 32]). On the other hand, while $\epsilon$-greedy exploration is explicitly unbiased in action space, policy churn likely prefers exploration across near-optimal actions (with respect to the current value function). This may be beneficial in some settings, for example when some actions are deadly, and high $\epsilon$ prevents long episodes. It may also be detrimental in others, where $\epsilon > 0$ helps the agent get unstuck.

## 3 Potential Causes

With the presence and impact of the churn phenomenon established, this section aims to provide additional depth. First, we look at the generality of the effect in Section 3.1. Second, we conduct investigations into the sensitivity of the phenomenon, with Section 3.2 showing ablations to the large-scale deep RL agents (more material in Appendix A), and Section 3.3 taking the complementary approach of interpolating between dynamic programming and a DQN approximation on a toy domain. Finally, Section 3.4 synthesises the findings and postulates some compatible underlying mechanisms.

### 3.1 Breadth of prevalence and non-causes

To judge the importance of the phenomenon, we need to establish whether it is specific to a narrow range of settings, or prevalent in a variety of domains, algorithmic variants, and hyper-parameters. It turns out that this is easy to do, because the effect is very much not a subtle one. In fact, policy churn is present in two very different deep RL agents, namely DoubleDQN [47] and (a variant

---

[2]In line with prior work, all our DoubleDQN experiments preserve the decaying $\epsilon$-schedule in the first $2\%$ of training, it is only zero after that initial phase; R2D2 experiments have no such schedule.

[3]Acting with the target network also introduces *latency* on how fast newly learned knowledge can be exploited. To see how specifically this latency should have a negligible effect on performance, imagine shifting the $x$-axis of the blue curve by 120 000 frames to the left, which would be an oracle "target-network-of-the-future" variant.

| Agent | DoubleDQN | R2D2 |
|---|---|---|
| Input | $84 \times 84$ grayscale | $210 \times 160$ RGB |
| Action set | minimal per game: $3 \leq |\mathcal{A}| \leq 18$ | full: $|\mathcal{A}| = 18$ |
| Reward | clipped | unclipped |
| Neural net | feed-forward, 1.7M parameters | recurrent, 5.5M parameters |
| Q-value head | regular | dueling |
| Update | 1-step double Q-learning | 5-step double Q-learning |
| Optimiser | RMSProp without momentum | Adam with momentum $= 0.9$ |
| Batch size | 32 | $32 \times 80 = 2560$ |
| Replay, replay ratio | uniform, 8 | prioritised (exponent 0.9), 1 |
| Parallel actors | 1 | 192 |
| Mean $\overline{\overline{W}}$ per update | $\approx 9\%$ | $\approx 6\%$ |

Table 1: The two agent setups considered differ in a number of properties. Given both settings result in similar churn, none of them seem critical. See Appendix B.1 and B.2 for details.

of) R2D2 [21], both widely used for training on Atari (see Appendix B.1 and B.2 for agent and environment specifications). Despite the large differences between the algorithms summarised in Table 1, the magnitude of policy change is surprisingly similar, indicating that it is unlikely that policy churn strongly depends on any of these specific choices.

The effect is also not specific to any environment: we measure similar magnitudes of policy change across a range of Atari games that vary in many dimensions, such as action space, reward scale and sparsity, deadliness, etc. Furthermore, it is present in all stages of training (see Figures 1 and 15). Unsurprisingly it is highest in early learning, but it remains high during training and even after evaluation performance converges or stabilises. [4]

### 3.2 Ablations

**Redundant actions.** A simple factor that could explain policy churn are redundant actions, i.e., when nominally different actions have the same outcome (in all, or a large fraction of state space). This property varies widely by environment (see also [31]), but we observe similar levels of policy change across them (Figures 1 and 15). Also, when exposing the full Atari action set in all games, creating explicit global redundancy, the churn magnitudes are not affected much (Figure 16). Appendix A.2 looks at the fine-grained aspect of *which* actions tend to be swapped for which others, and finds no structure easily related to the (known) equivalence relations (Figure 10). In other words, most $\arg\max$ changes are *not* happening between equivalent actions.

**Small action gaps.** Another potential factor for large policy churn with greedy policies can be the interplay between FA and small action gaps (difference between largest and second-largest action values): small approximation error can suffice for sub-optimal actions to overtake optimal ones. This hypothesis predicts that value learning methods inducing larger action gaps (e.g., Advantage Learning (AL, [3]) which artificially lowers the values of sub-optimal actions) could reduce policy churn. Figure 3 (right) shows that indeed policy churn is decreased substantially by AL, correlating with an increase of action gaps (which consistently grows under AL, see Figure 17). Curiously, this does not seem to severely diminish the remaining policy churn's effectiveness for exploration: Figure 18 shows successful training of an AL-DQN with $\epsilon = 0$.

**Non-stationary state distribution.** Another explanatory hypothesis for the observed large magnitude of policy churn relates it to the non-stationary data distribution caused by the policy generating the data, which is evolving. Even when the agent converges to near-optimal performance, like in PONG, it may happen that the policy keeps changing on states where actions are inconsequential (where multiple actions have near-equal value), causing non-stationarity in the data distribution and thereby driving further policy churn. To test this, we utilize the "forked tandem" setting from [34], in which a high-performing policy is trained on the stationary data distribution generated by its initial

---

[4]Given the high churn in (preliminary) experiments on DM-Lab [1], as shown in Figure 25, we also consider it unlikely that the phenomenon is specific to the Atari setting.

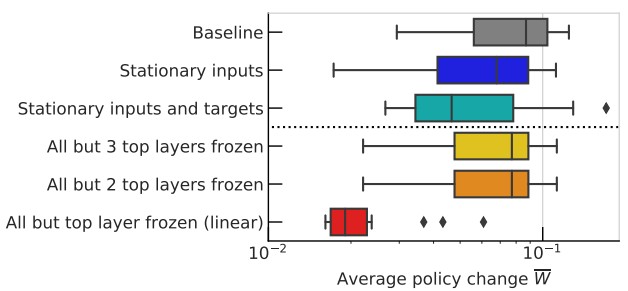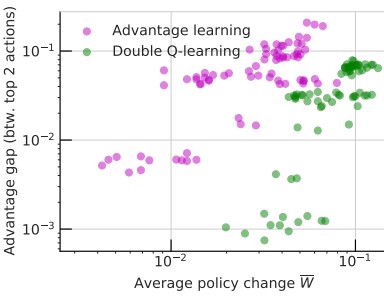

Figure 3: **Left:** Ablations in the "forked" DoubleDQN setting (5 games, 3 seeds each). Training progresses as normal until 50M frames, after which either the data distribution is fixed ("stationary"), or part of the neural network is frozen; "stationary targets" denotes regression onto Monte-Carlo returns. The only large effect on policy change is when switching from deep learning to linear FA (**red**). For more detail see Figures 19, 20, and 22. **Right:** Relation between policy change and advantage gaps. Different points correspond to different games, seeds and stages of learning. This shows that the action-gap-increasing algorithm (**purple**) reduces policy change; and also how, given an algorithm, advantage gaps correlate with churn. For more detail see Figures 17 and 18.

snapshot (see below for more details). As can be seen in Figures 3 (left) and 19, the high level of policy churn is still preserved in this stationary-data regime, ruling out data non-stationarity as the main driver of the phenomenon.

**Non-stationary targets.** Temporal difference (TD) learning can give rise to another form of non-stationarity, as the bootstrap targets change at the pace of the target value function, an aspect that is preserved even in the forked tandem setting. To sidestep this, a simple control experiment uses the same setup but with Monte-Carlo returns as learning targets, turning it into pure policy evaluation via supervised regression (with noisy targets). The results (Figures 3, left, and 20) show a similar level of churn to the Q-learning updates, indicating that the phenomenon is not specific to TD-based algorithms.

**Decoupled acting and target networks.** To assess *how much* policy churn is beneficial for exploration, we devise the following setup: A greedy policy based on the agent's target network drives behaviour, and we vary the frequency with which it is synchronized to the online network. To avoid conflation with an adverse effect on learning stability, we keep the regular target network update frequency constant, while using an additional *acting network*, periodically copied from the online network, for behaviour generation alone. Figure 21 shows that an acting network updated more than every $\approx 1\,000$ gradient updates achieves most of the exploration benefits of acting with the online network, though in some games higher frequencies yield further benefit. This implies that the amount of policy change needed for exploration is much smaller than what is generated by DQN's learning process (otherwise the observed collapse would happen at much smaller update intervals).

**Self-correction and the tandem effect.** In [34] a phenomenon dubbed the "tandem effect" was observed: the failure of a deep RL agent to adequately learn from the training data generated by a different instance of the same agent, highlighting the importance of *self-correction* by interactively generated data from the policy being trained. One of their settings, the "forked tandem", starts with a copy of a high-performing policy and uses data sampled from its stationary distribution to continue training; even this apparently benign scenario leads to instability and potential collapse of the trained policy. Policy churn may provide a partial mechanistic explanation for the origin of the instability, showing that rapid policy change can be expected at all stages of training. The observation that the trained policy changes on a significant proportion of states at every update (performed on a negligibly small sample, a single minibatch of 32 state transitions) supports the hypothesis from [34] that erroneous extrapolation or over-generalization may play a key role in causing deviation and instability and producing the tandem effect in the absence of corrective training signal from self-generated data. Analogously to results in [34] we observe that the magnitude of policy churn is highly correlated with the depth of the trained function approximator, further supporting this hypothesis; see Figures 3 (left) and 22, which show specifically how policy change drops dramatically in a linear FA regime.

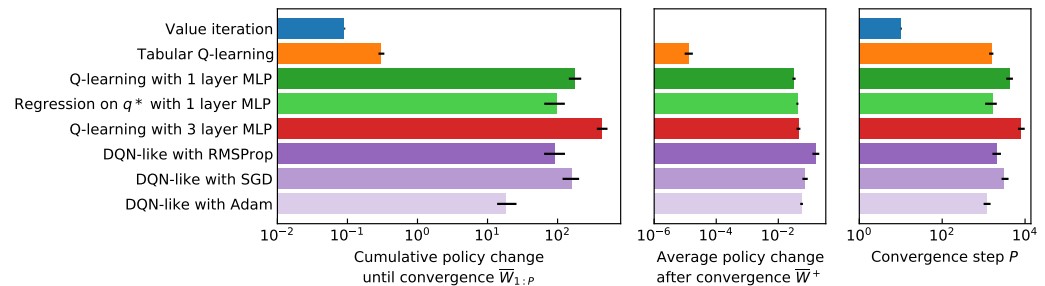

Figure 4: Ablations of what aspects drive policy change in CATCH. Unless specified each FA variant uses the SGD optimiser. Here "DQN-like" is "Q-learning with 3 layer MLP" with replay and mini-batches. Error bars denote the interquartile range over 100 seeds.

## 3.3 Detailed case study: CATCH

CATCH is a toy environment where the total number of states is small enough to be amenable to ground-truth dynamic programming approaches using the explicit matrix of transition probabilities, while providing an observation space that requires non-linear FA to represent the optimal value function. We construct a spectrum of settings that all learn the optimal policy after some $P$ iterations. For each we measure cumulative policy change $\overline{W}_{0:P}$ and average change after convergence $\overline{W}^{+}$ (Eq. 3). Main results are shown in Figure 4. Sitting at one end of the spectrum, exact value iteration converges after $P = 10$ steps with $\overline{W}_{0:P} = 0.09$ (because the initial policy is random, and the optimal policy has ties in most states). The next simplest setting is tabular Q-learning with incremental updates (here and elsewhere, hyperparameters like the learning rate are tuned for fast convergence to optimal performance, see Appendix B.3 for details). The next steps add in non-linear FA (shallow with 1 hidden layers, then deep, with 3). At the other end of the spectrum is an approximation to DQN which includes experience replay, mini-batches, a deep neural network, and an advanced optimiser (RMSProp [46]). Figure 4 also shows intermediate variations such as supervised regression to $q^{*}$, and different optimisers (e.g., Adam [23]); additional results in Appendix A.1. Overall, the experiments in this section show that, among the factors we considered, the presence of function approximation is by far the aspect that correlates the most with the occurrence of the phenomenon of policy churn.

## 3.4 Mechanistic hypothesis

In an attempt to synthesise all the evidence presented so far, we propose that the observed policy churn is primarily a result of two components that need to be present jointly:

1. Non-linear, global function approximation (such as deep neural networks), where each update can affect all states and all actions.
2. A learning process with a high amount of noise. This could have multiple sources: stochastic optimisation (e.g. small batch sizes, large learning rates), noisy learning targets, non-stationary data (or targets, as with bootstrapping), an imbalanced data distribution (e.g., seeing some actions more often than others).

In other words, the rapid policy change dubbed churn is the symptom of high-variance updates to a global function approximator, and neither aspect in isolation is enough to promote it.[5]

---

[5]A compounding effect could be a *mismatch* between the regression loss in value space that drives the learning process, and policy space, which is what matters for performance and for exploration (and is measured here). While supervised learning does not have this mismatch, it could conceivably have large "policy change" too. An in-depth treatment exceeds the scope for this paper, but preliminary results on MNIST (Appendix B.5) indicate *no* large churn in the supervised setting.

# 4 Where do we go from here?

## 4.1 Learning at the edge of chaos

Deep learning has a well-known trade-off between speed of learning and stability that incentivises tuning the learning dynamics to be near the edge of chaos[6] [10]. The presence of policy churn could enrich this picture in two ways for the case of deep RL. First, to the extent that rapid policy change helps drive exploration (Section 2), this is an additional incentive to keep learning dynamics sufficiently noisy. Second, to the extent that value-based learning relies on self-correction [34], that is, the actions to be corrected need to be picked by a greedy policy, this also encourages a large amount of policy change. Circumstantial evidence for these is that value-based RL tends to require additional stabilisation mechanisms (e.g., target networks), less common in policy-based RL.

## 4.2 Policy null-spaces

Policy change is necessary for policy improvement, but does not necessarily imply change in performance. We define the space of policies that have the same value as a reference policy $\pi$ (in all states) as its *null space* $\text{Null}(\pi)$. One way to quantify it is the *diameter*

$$|\text{Null}(\pi)| := \max_{\pi', \pi'' \in \text{Null}(\pi)} \overline{W}(\pi', \pi'')$$

as the largest policy change possible between any pair of policies within it. Typical scenarios with large null spaces have states in which actions have no effect (e.g., move actions while falling), multiple actions with the same effect (globally, or locally in part of state space), or multiple paths leading to the same outcome (e.g., up-then-left vs. left-then-up). These scenarios are common among the environments typically studied in deep RL. Appreciating that policy null spaces can be large makes the policy churn phenomenon more palatable, helping us reconcile that agents changing their mind a million times about the best action (on average in each state) can have excellent performance. A particularly interesting null space is the one around the optimal policy $\text{Null}(\pi^*)$: when it is large, there are many "safe" ways to keep changing the policy (possibly by a lot) after converging to maximal performance, which is what we observe on, e.g., PONG (see Figure 14), at least when $\epsilon > 0$.

## 4.3 Off-policy corrections in the presence of churn

Most types of off-policy correction are based on the gap between the data-generating behaviour policy $\mu$, and a target policy $\pi$. When doing multi-step updates, conservative methods truncate trajectories to bootstrap early compared to on-policy experience [25]. To obtain the benefit of multi-step value propagation, the average truncation length cannot be too short, and thus a silent assumption is that $\mu$ and $\pi$ do not differ too frequently. Empirically however, multi-step back-ups without *any* off-policy correction can be surprisingly effective [20]. We can now re-interpret these findings through the lens of policy churn: as the greedy policy changes much faster than expected, even a slight latency (a few updates) between the parameters of the data-generating policy and the current target policy leads to massive truncation effect. If consecutive greedy policies are (approximately) within the null-space of each other, the benefits of an uncorrected multi-step update may outweigh its cost. It also suggests the possibility of new off-policy algorithms that exploit the knowledge of the churn phenomenon, by truncating in a less aggressive fashion, motivated by the intuition that majority of rapidly changing policies lie within an (approximate) null space of each other (see Appendix A.5).

## 4.4 The social dynamics of research

If policy churn is indeed a hidden form of exploration, how did it come about? It seems unlikely that a useful mechanism emerges completely by chance. An intriguing possibility is that policy churn is the effect of a gradual process of *natural selection* [12]. The hypothesis is that, if policy churn provides benefits, algorithms that display some level of it would be favoured over their counterparts in the inevitable engineering work surrounding the design of large-scale agents. In this view, RL practitioners play the role of "nature" exerting a selective pressure that shapes algorithms across time. This process could be completely *unconscious* to the researchers involved: agents under-performing

---

[6]As epitomised in the heuristic to tune the learning rate to the largest value that does not explode.

due to weak exploration would be discarded in favour of agents that explored better using some form of (hidden) policy churn. Over time, the multiple degrees of freedom of large-scale agents (hyper-parameters, network architecture, etc.) would be tuned to reflect just the right amount of churn: enough to help in exploration, but not too much to make the overall learning process unstable.

As appealing as the above hypothesis may be, as of now we do not have any evidence to support it. Still, it is worth considering, as it raises many interesting questions. Are there other hidden effects that have been selected for over the years? How many good design choices got discarded because they happened not to promote policy churn (or other similarly hidden effects)? Is the AI research community narrowly focused on a handful of design templates that happen to induce some ill-understood dynamics? This sort of question did not arise in the past, when agents were simple enough that we could keep track of their functioning at the finest level of detail. Now that deep RL agents have reached a certain level of complexity, any design choice may have a cascade effect whose consequences we do not anticipate. This creates the perfect environment for the sort of selective process described above. Acknowledging this possibility and being aware of it may be an important step toward unveiling hidden effects, and perhaps turning them into more purposeful design.

## 5 Related work

A phenomenon in the literature that is related to policy churn, but not equivalent to it, is that of *policy oscillation* or *chattering* [16, 4, 48]. Bertsekas and Tsitsiklis [4] define the *greedy region* of a function $q \in \mathcal{Q}$ as the set of functions in $\mathcal{Q}$ that induce the same greedy policy as $q$. Each greedy region has an associated "fixed point": the value function of the greedy policy induced by the functions in it. Conversely, every value function is the fixed point of a greedy region. It is well known that the only value function that belongs to its own greedy region is the optimal value function $q^*$ [4]. However, when function approximation is used, projecting value functions onto the FA space may create cycles that repeat indefinitely, a phenomenon known as policy oscillation. We do not think policy oscillation is a likely explanation for policy churn, because the approximators used in our experiments should have the capacity to approximate any value function to a reasonable level of accuracy. This is corroborated by Figures 3 and 4 and Appendix A.1, which show neural networks with more or wider layers exhibit *more* churn. We believe it is more instructive to think of each update of the approximator as moving the value function across the boundaries of greedy regions. As discussed, in many cases this has no effect on the agent's performance, since the policies associated with neighbouring greedy regions may belong to each other's null space (Section 4.2).

Another body of related work is the literature on stochastic gradient descent, and how a relatively high amount of stochasticity can be beneficial to optimisation by overcoming local optima and converging to flatter optima with better generalisation properties [22, 37, 24]; this is such a prominent effect that in some cases it is even beneficial to inject additional noise with Langevin dynamics [50]. More loosely related are studies of learning in animals, exhibiting large drift in synaptic or representation space [30, 38, 26, 13, 42], as well as evidence for highly variable behaviour policies that can get consolidated through salient dopamine events [9].

## 6 Conclusions and Future Work

**Revisiting interpretations.** Nine years after the introduction of DQN [27], there are still phenomena in value-based deep RL that remain to be understood, with this paper putting the spotlight on one of them that lies at the intersection of learning and exploration dynamics. In particular, we hope that an awareness of the churn phenomenon will make researchers revisit some good ideas that may have been prematurely disregarded or under-valued, either because their promised exploration effect was too entangled with learning dynamics and the resulting churn, or because they improved the stability of learning dynamics at the expense of reduced exploration that undid the overall gains.

**Churn beyond value-based RL.** An obvious follow-up question is whether policy churn is also an important effect beyond value-based algorithms. We could imagine that actor-critic algorithms incur much less policy change, because stochastic policies change more smoothly, or because various penalty terms keep the updated policy from deviating too much from its precursor. If that were the case, it would indicate that exploration in these agents differs as well, possibly with complementary advantages and disadvantages. Similarly, various instances of model-based RL [29, 43] may have

less policy change, because the planning process could mitigate some of the function approximation effects. We leave these investigations to future work, but include one preliminary result (Figure 26) that seems to hint at churn being high in some actor-critic agents as well.

**Explicit and controllable churn.** If policy churn is indeed a valuable and non-trivial exploration mechanism, then it may be costly to deliberately abandon it, especially if its effect turns out to be complementary to simpler noise-based exploration mechanisms. Ideally we would want an explicit and controllable mechanism that produces the same kind of consistent, non-harmful exploration behaviour, but without such a tight entanglement with the learning dynamics. The core benefit of such a division of labour would be that practitioners could study, change or tune the learning and exploration processes separately, without having to make arbitrary trade-offs (e.g., around stability, diversity, representations) that inevitably arise when they are considered in combination.

## Acknowledgments

The ideas presented here were refined in discussion with numerous of our DeepMind colleagues. Will Dabney, Joseph Modayil and Matteo Hessel helped improve the paper with detailed feedback, and we thank David Silver, Diana Borsa, Miruna Pîslar, Claudia Clopath, Vlad Mnih, Iurii Kemaev, Junhyuk Oh, Bilal Piot, Greg Farquhar, Dan Calian, Hado van Hasselt and Alex Pritzel for their input. We also thank the anonymous NeurIPS reviewers for their suggestions, as well as the many RLDM and ICML attendees whom we put our guesstimate survey question to.

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
