# A    Additional Results and Ablations

This appendix contains a number of figures that are already referenced from within the main paper.

## A.1    Additional results on CATCH

Complementing Figure 4 in Section 3.3 are Figures 5, 6, and 7 which show policy change for additional variants, in particular wider networks, behavioural cloning of $\pi^*$, and learning rate annealing.

**Policy change per state.** Figure 8 shows the policy change per state averaged over different periods of training and $1\,000$ seeds for the "DQN-like" agent with RMSProp optimiser. See Appendix B.3 for exact hyper-parameters. Note that episodes in CATCH always start with the paddle in the centre. This means some states shown in the plots in Figure 8 are not actually possible, in particular states corresponding to the dark top row of cells in all but the central column of plots. Another consequence is that starting states (corresponding to the top row of cells in the middle column of plots) have disproportionately more policy change. Indeed, in a version of the environment where the paddle is initialised randomly, this large relative difference in policy change disappears. After convergence, in states where the action gap is high (states where the ball is diagonal from the paddle) there is little policy change as expected and most of the policy change happens in states corresponding to the ball higher up where the exact actions taken matter less; see also Figure 9. There is some conflation from the state distribution induced by the policy; everything else being equal, policy change is higher in states that are updated more often. At the start of training and even a little while after convergence, for states where the paddle is on one of the sides (the first and last column of plots) and the ball is directly just above the paddle the relative amount of policy change is low. But well after convergence this flips and policy change is relatively high in these states. Presumably this is because early in training the agent has yet to learn that values for no-op action and the action that would move the paddle into the wall have the same effect.

## A.2    Redundant action spaces

The DoubleDQN and R2D2 settings differ in the actions spaces used to act in the set of Atari games. As indicated in Table 1, DoubleDQN always employs the minimal action set $3 \le |\mathcal{A}| \le 18$ (see subplot titles in Figure 15), while R2D2 always uses the full action set $|\mathcal{A}| = 18$. Adding to that, the experiments in Figure 10 also include a "redundant $\times 3$" setting where the full action set is artificially replicated 3 times ($|\mathcal{A}| = 54$).

## A.3    Unlimited policy change in a two-armed bandit

One minimalist setting in which it is possible to obtain large (cumulative) policy change is incremental learning of similar Q-values using small step-sizes. For example, consider learning the two (tabular) Q-values of a two-armed bandit. Q-values are initialised near each other ($q_0(a_1) \approx q_0(a_2)$), and their true targets are also nearly identical ($q_P(a_1) \approx q_P(a_2)$, but far from initialisation, $q_0(\cdot) \ll q_P(\cdot)$). With that set-up, a learning process that alternates between the actions to update can produce an $\arg\max$ switch on each update, because the last-updated Q-value will always be the larger one of the

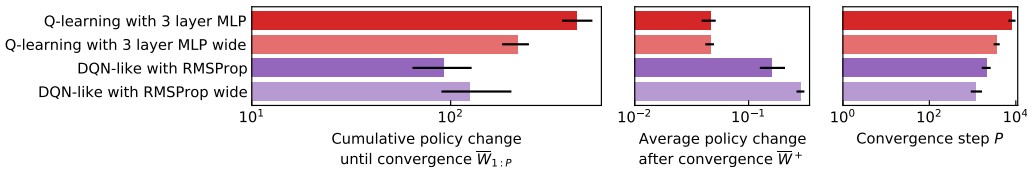

Figure 5: Ablation on the **width** of the network in CATCH. Here "wide" means the neural network has 200 units per hidden layer instead of 50. Increasing the width of the network increases policy change metrics $\overline{W}_{1:P}$ and $\overline{W}^+$ in the DQN-like variant whereas for the Q-learning with 3 layer MLP variant, it is the opposite.

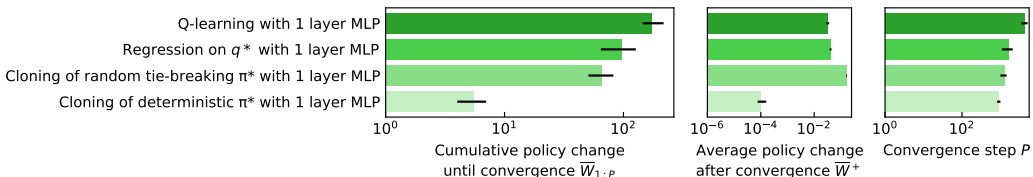

Figure 6: Additional supervised variants of CATCH: behavioural cloning of $\pi^*$ with a cross-entropy loss and ground-truth targets. Note that the high $\overline{W}^+$ in the random tie-breaking variant is due to the many exact ties at the optimum; the deterministic variant has low $\overline{W}^+$.

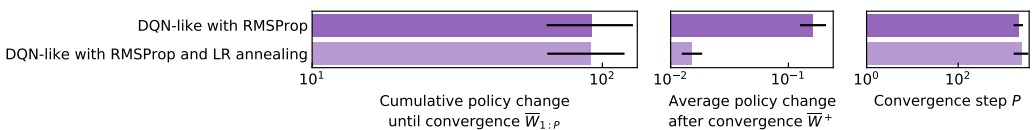

Figure 7: Variant on CATCH where the learning rate is annealed from $10^{-3}$ to $10^{-4}$ over $10\,000$ steps. As expected the resulting average policy change after convergence is lower.

two. And with the appropriate setting of step-sizes and initialisation, $P$ and thus $\overline{W}_{1:P}$ can be made arbitrarily large.

## A.4 High policy change in dynamic programming

Throughout the paper we treated policy change as an unexpected phenomenon. However, some amount of policy change is inherent to all RL algorithms. Value-based methods, in particular, are based on dynamic programming, which has at its core two operations: policy evaluation and policy improvement. Since by definition policy improvement involves change, it is fair to ask: how *much* change is in fact expected? In other words: if we could isolate all other effects, like approximation and noise, how much policy change would still remain?

In Section 3.3 we already touched on this subject with the experiments on CATCH using value iteration. In this section we revisit the question and try to provide a more definite answer to it. As it turns out, and perhaps not surprisingly, the answer to this question seems to be very domain dependent. The expected amount of policy change that is inherent to dynamic programming can vary significantly from one environment to the other.

To illustrate this point, we now describe a simple policy evaluation setting that does not involve any approximation, incremental learning, or noise; and yet we see a large amount of policy change happening. Given the value function $q_\pi$ of a policy $\pi$, we compute the greedy policy $\pi'$ with respect to $q_\pi$, and monitor the changes in the greedy policy induced by the intermediate functions as we move from $q_\pi$ to $q_{\pi'}$.

To describe our example precisely, we will need two concepts. First, we define the *greedy operator* $g : \mathcal{Q} \to \Pi$ as

$$g(q) = \pi \text{ such that } \pi(s) = \mathrm{argmax}_a q(s, a), \text{ for all } s \in \mathcal{S},$$

where $q$ is an arbitrary function in $\mathcal{Q}$ and ties are broken in an arbitrary, but consistent, way. It will also be convenient to introduce the *Bellman operator* of a policy $\pi$ as

$$T_\pi q(s, a) = r(s, a) + \gamma \mathbb{E}_{S' \sim p(\cdot|s,a), A' \sim \pi(\cdot|S')} \left[ q(S', A') \right],$$

where $q \in \mathcal{Q}$, $r(s, a)$ is the expected reward following the execution of $a$ in $s$, $p(s'|s, a)$ is the probability of transitioning to state $s'$ given that action $a$ was executed in state $s$, and $\mathbb{E}[\cdot]$ is the expectation operator. It is well known that $\lim_{k \to \infty} T_\pi^k q = q_\pi$ for any $q \in \mathcal{Q}$.

Equipped with the concepts above, we can now present our example. Figure 11 shows an MDP composed of an arbitrary number of states structured as two chains. We are interested in monitoring how the policy will change in state $s$ as we do policy evaluation. Suppose that we start with a policy

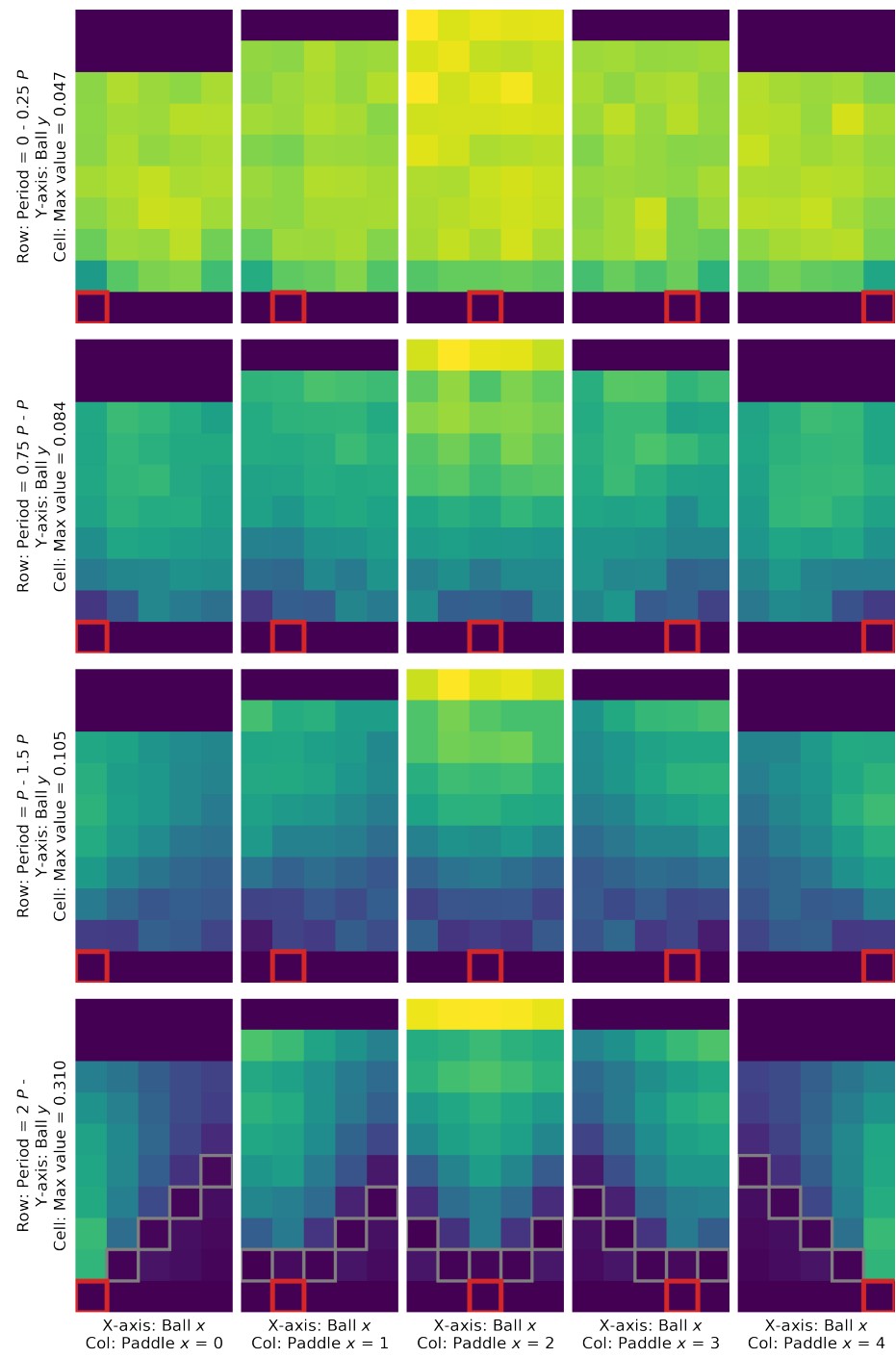

Figure 8: Policy change **per state** averaged over different periods of training and seeds, on CATCH. Each row of subplots represents a given period of training expressed in multiples of $P$ (performance convergence step), namely "early", "pre-convergence", "post-convergence" and "late". Each column corresponds to states where the paddle is at a particular $x$-coordinate, also highlighted by a red square. Each cell on a given figure represents the state corresponding to the $(x, y)$ position of the ball. The subplots in each row share the same scale, from 0 to a max value indicated on the $y$-axis label. Cells highlighted by a grey square correspond to states where the action gap is non-zero for $q^*$, and we see that policy change is indeed lowest there, after convergence. See also Figure 9.

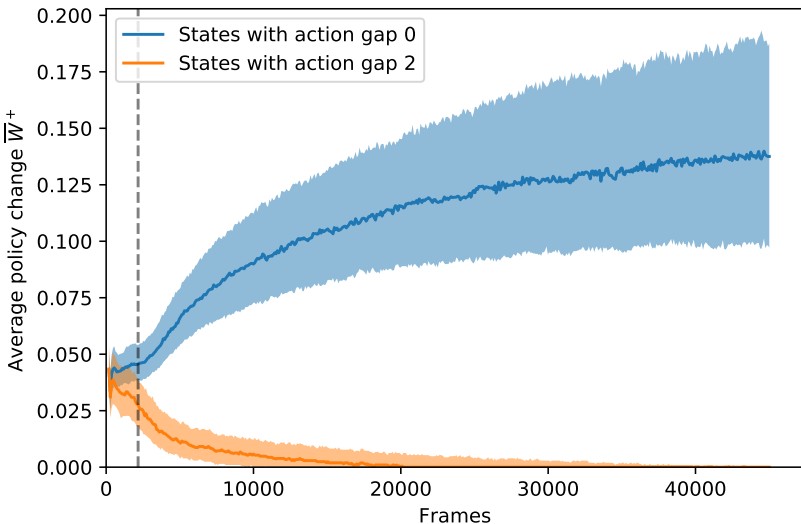

Figure 9: Policy change across two groups of states in CATCH with the DQN-like agent with RMSProp. The grey dotted line is the median convergence step $P$. Shaded areas denote the inter-quartile range over $1000$ seeds. This shows how late in training, and well after convergence, policy change concentrates in the null space (where action gaps are zero, blue) and critical actions (orange) are perturbed less and less often. See also Figure 8.

$\pi$ that selects action **red** everywhere. Clearly, $v_\pi(s) = 0$. The greedy policy $\pi' = g(q_\pi)$ will select actions associated with nonzero rewards whenever they are available; when they are not available, we will assume that the greedy operator $g$ will resolve the ties by always picking the **green** or the **blue** action over their **red** counterpart.

Starting from $q_\pi$, we will now monitor how much the greedy policy $g(T_{\pi'}^k q_\pi)$ changes in $s$ with the sequence $k = 1, 2, ...$, that is, as we move from $q_\pi$ to $q_{\pi'}$ as part of policy evaluation. For ease of exposition, we will use $\pi_k \equiv g(T_{\pi'}^k q_\pi)$ to refer to the greedy policies along the way. Clearly, in the first step, when we change from $\pi$ to $\pi_1 = g(T_{\pi'}^1 q_\pi)$, the policy changes in $s$ from **red** to **green**. Now, in the second step, an easy calculation shows that the policy changes again, now from **green** to **blue**. If we keep doing this exercise, a simple pattern emerges: policies $\pi_k$ whose index $k$ is odd will pick action **green** in $s$, while their counterparts with an even index will instead select **blue** on that state. This means that $W(\pi_k, \pi_{k+1}|s) = 1$ along the sequence of greedy policies $\pi_1, \pi_2, ...$.

This deliberately simple example illustrates that the maximum possible amount of policy change can happen on a given state simply as an effect of policy evaluation. It is not difficult to construct examples in which a similar effect is observed throughout state space.

In Section 5 we discussed how the well-known policy oscillation effect may be responsible for part of the policy change when function approximation is used. The "dynamic-programming effect" discussed in this section happens in addition to that, regardless of function approximation. In general, we expect that policy change could be a result of both effects, plus other causes like the ones discussed in Section 3.4 and Appendix A.3. Given all the empirical evidence we have collected, we are reasonably confident that the causes discussed in Section 3.4—namely, global function approximation and noise—play a much more important role than the policy oscillation and dynamic programming effects in the setup studied.

## A.5 Churn-aware off-policy correction

Following up on Section 4.3, this section spells out some concrete possibilities for forms of off-policy correction that take the churn phenomenon into account. In a low-latency setting for example, it may be worth truncating traces when the noise of $\epsilon$-greedy leads to a low-advantage action getting executed, but not when the action discrepancy is purely due to churn ($\mu$ was acting greedily). Alternatively,

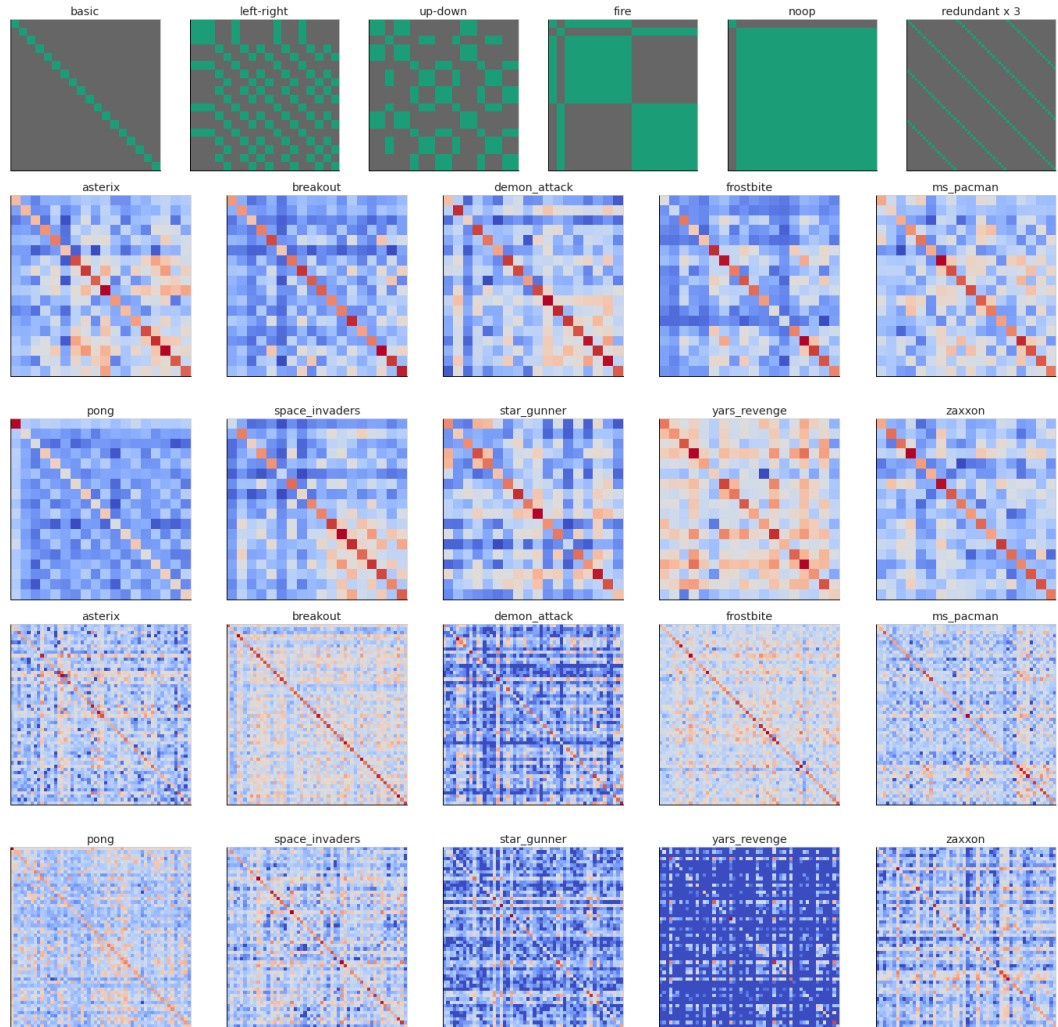

Figure 10: Confusion matrices: between which actions do the $\arg\max$ switches happen? **Top row**: patterns that we could expect to see in games where all actions are distinct ("basic"), where only left-right movement matters ("left-right"), etc. **Middle rows**: empirical confusion statistics from an R2D2 experiment, warmer colors indicate higher likelihood (log-scaled). Note that some games have an effectively reduced action set; for example, in PONG only up/down/no-op matters, but this pattern ('up-down') does not show up in the switch statistics. **Bottom rows**: empirical confusion statistics in an ablation experiment where all actions were redundantly replicated three times (unbeknownst to the agent): here we would expect a pattern to emerge like in the top right ("redundant ×3") if the agent were to find out about the redundancy and only switch between these; but this does not happen.

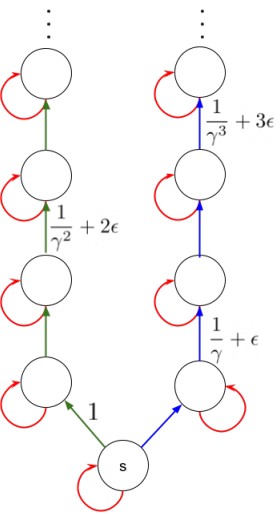

Figure 11: Example of MDP in which a lot of policy change can happen at state $s$ during exact policy evaluation. States are represented as circles and actions are represented as arrows. Rewards are zero in all transitions except when marked otherwise in the diagram.

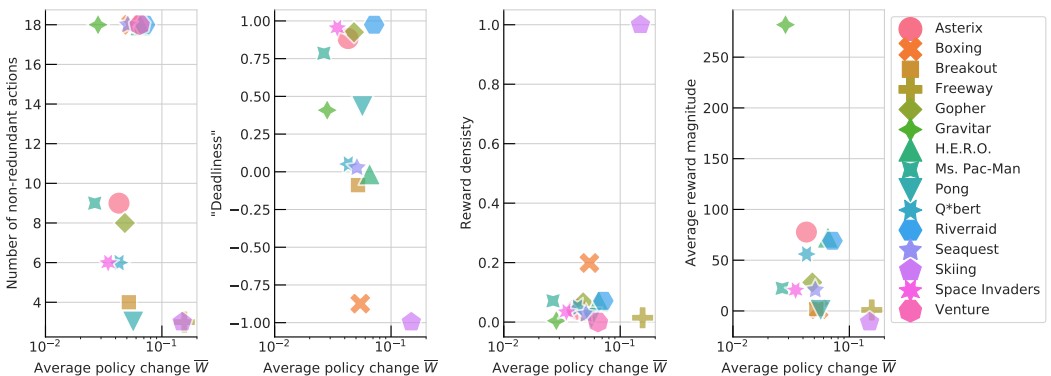

Figure 12: Relating average policy change $\overline{W}$ (across all seeds and time periods, in the R2D2 setup) to various game-specific properties. The number of non-redundant actions refers to the minimal action set (used directly in DoubleDQN). "Deadliness" is the correlation between episodic return and episode length. Reward density is the fraction of transitions that produce a non-zero reward, and the average reward magnitude is the average return divided by the number of non-zero reward events.

we think it is plausible to make truncation decisions based on (relative) advantage gaps, effectively ignoring $\arg\max$ switches between actions of similar value.

### A.6 Relating churn to other game-specific properties

Overall, we have not identified game-specific properties that are clearly predictive of the magnitude of policy change. Figure 12 provides a number of scatter plots for game-specific properties that we had considered as possibly having an influence.

## B Experimental Details

### B.1 DQN experiments

We chose to use double Q-learning with DQN (DoubleDQN, [47]) instead of vanilla DQN [27, 28] for all of our experiments, as it is generally the more robust and better tuned of the two algorithms. Apart

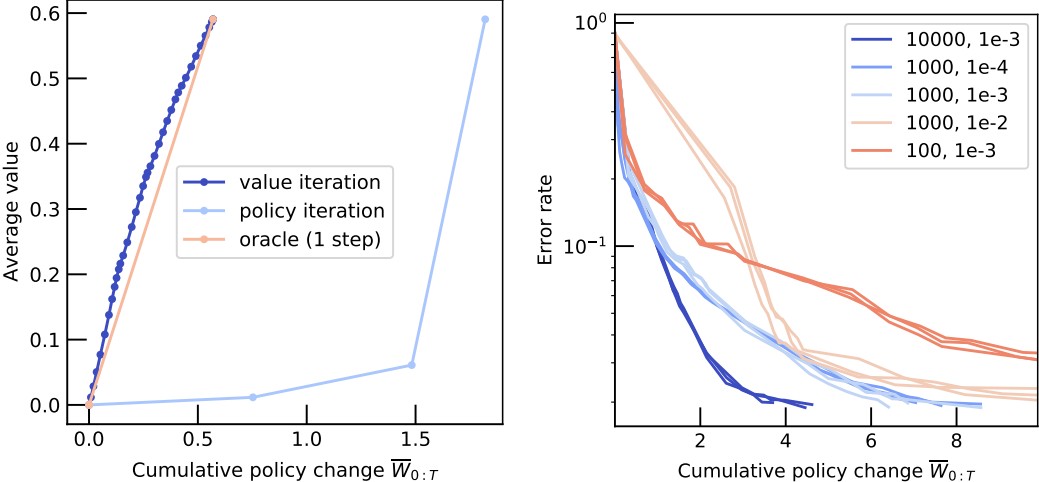

Figure 13: Performance as a function of total accumulated policy change $\overline{W}_{1:T}$. **Left:** Simple dynamic programming baselines in a tabular grid world. In this scenario, value iteration (blue) goes through $P = 37$ steps until reaching $q^*$, but does not accumulate more policy change ($\overline{W}_{1:P} = 0.57$) than an oracle that jumps from $q_0$ to $q^*$ (pink). Policy iteration does just $P = 3$ steps but accumulates $\overline{W}_{1:P} = 1.82$. Each iteration or update is shown as a dot. **Right:** Supervised training of an MLP on MNIST with various hyper-parameter settings, listed as (batch size, learning rate) pairs in the legend. The multiple lines correspond to 3 different random seeds for each setting. Overall, vanilla MNIST training goes through a handful of label changes per input (on average) over the course of training.

from overall improved performance, for the purposes of this investigation there is little difference between the two, notably in terms of policy change, see Figure 23. We use an identical setting as the original DoubleDQN paper, including all hyper-parameters (which differ slightly from those in vanilla DQN). The main ones are listed in Table 1, the remaining ones in Table 2. Our implementation is based on a slightly modified variant of the open-source DoubleDQN implementation in DQN Zoo [36].

Our Atari investigations did not involve any hyper-parameter tuning. The modifications we did to existing settings for the exploration experiments (Section 2) are binary ablations:

- Reducing $\epsilon = 0.01$ to $\epsilon = 0$ in the $\epsilon$-greedy behaviour policies
- Using the target network instead of the online network for acting.

The "forked tandem" setup used in several ablations in Section 3.2 follows [34] and is based on their accompanying open-source implementation.[7]

Our Atari experiments are run with the same ALE variant of the Atari 2600 benchmark [2] as in the original DQN and DoubleDQN works, using an action repeat of 4, a zero discount on transitions involving a life loss, and the only source of stochasticity being a random number (uniformly between 0 and 30) of no-op actions applied at the beginning of each episode. Unless stated otherwise, all these experiments are run with 3 seeds for each configuration.

A lot of preliminary investigations used a small subset of Atari games (BREAKOUT, PONG, MS. PAC-MAN and SPACE INVADERS). For the final runs on 15 games, we picked a representative subset of the 57 Atari games, with a preference for games on which DoubleDQN can achieve a decent performance level.

### B.2 R2D2 experiments

The agent denoted as "R2D2" throughout the paper is a variant of the Recurrent Replay Distributed DQN architecture [21]. It comprises 192 CPU-based actors concurrently generating experience

---

[7]`https://github.com/deepmind/deepmind-research/tree/master/tandem_dqn`

| Agent | DoubleDQN | R2D2 |
|---|---|---|
| Convolutional torso channels | $32, 64, 64$ | $32, 64, 128, 128$ |
| Convolutional torso kernel sizes | $8, 4, 3$ | $7, 5, 5, 3$ |
| Convolutional torso strides | $4, 2, 1$ | $4, 2, 2, 1$ |
| Pre-LSTM linear layer units | N/A | 512 |
| LSTM hidden units | N/A | 512 |
| Post-LSTM linear layer units | N/A | 256 |
| Value head units | 512 | Dueling $2 \times 256$ |
| Action repeats | 4 | 4 |
| Actor parameter update interval | 4 steps | 400 steps |
| $\epsilon$ for $\epsilon$-greedy policy | annealed from 1 to 0.01 | fixed 0.01 |
| Replay sequence length | 1 | 80 |
| Replay buffer size | $10^6$ | $4 \times 10^6$ observations |
| Priority exponent | N/A | 0.9 |
| Importance sampling exponent | N/A | 0.6 |
| Discount $\gamma$ | 0.99 | 0.997 |
| Target network update interval | $120\,000$ frames ($7\,500$ updates) | 400 updates |
| Gradient clipping | $\frac{1}{32}$ | N/A |
| Normalisation | N/A | Return-based [40] |
| Optimiser & settings | RMSProp [46] | Adam [23], |
| | learning rate $\eta = 2.5 \times 10^{-4}$, | learning rate $\eta = 2 \times 10^{-4}$, |
| | decay $= 0.95$, $\epsilon = 10^{-6}$ | $\beta_1 = 0.9$, $\beta_2 = 0.999$, $\epsilon = 10^{-6}$ |

Table 2: Atari agent hyper-parameter values (in addition to those in Table 1). These follow [47] and [40], respectively.

and feeding it to a distributed experience replay buffer, and a single GPU-based learner randomly sampling mini-batches of experience sequences from replay and performing updates of the recurrent value function by gradient descent. The value function is represented by a convolutional torso feeding into a linear layer, followed by a recurrent LSTM core, whose output is processed by a further linear layer before finally being output via a Dueling value head [49]. The exact parameterization follows the slightly modified R2D2 presented in [11, 40], see Table 2 for a full list of hyper-parameters. It is trained using the Adam optimiser [23] on a 5-step Q-learning loss, using a periodically updated target network for bootstrap target computation. Replay sampling is performed using prioritized experience replay [41] with priorities computed from sequences' TD errors following the scheme introduced in [21]. The agent uses a fixed replay ratio of 1, i.e. the learner or actors are throttled dynamically if the average number of times a sample gets replayed exceeds or falls below this value. It also uses unclipped rewards and unclipped gradients, and an accompanying return-based normalisation, as in [40]. Differently from those Atari RL agents following DQN [28], our agent uses the raw $210 \times 160$ RGB frames as input to its value function (one at a time, without frame stacking), though it still applies a max-pool operation over the most recent 2 frames to mitigate flickering inherent to the Atari simulator. As in most past work, an action-repeat of 4 is applied, episodes begin with a random number of no-op actions (up to 30) being applied, and time-out after $108\,000$ frames (i.e. 30 minutes of real-time game play). The agent is implemented with JAX [5], uses the Haiku [17], Optax [7], Chex [6], and RLax [18] libraries for neural networks, optimisation, testing, and RL losses, respectively, and Reverb [8] for distributed experience replay.

All our experiments ran for $40\,000$ learner updates. With a replay ratio of 1, sequence length of 80 (adjacent sequences overlapping by 40 observations), a batch size of 32, and an action-repeat of 4 this corresponds to a training budget of $\approx$ 200M environment frames ($\approx$ 100 times fewer than the original R2D2). In wall-clock-time, one such experiment takes about 2 hours. All experiments are conducted across 15 games, using 3 seeds per game, unless stated otherwise.

## B.3 CATCH experiments

For CATCH [33] experiments, Table 3 lists the hyper-parameters for each of the variants specified in Figure 4. For each variant, seeds that did not converge after $5\,000$ episodes of training were filtered out. In practice, all seeds for all variants in the table converged. For all CATCH experiments

| Value iteration | | |
|---|---|---|
| Tabular Q-learning | Learning rate | 0.1 |
| | Batch size | 1 |
| Q-learning with 1 layer MLP | Learning rate | 0.1 |
| | Batch size | 1 |
| | Optimiser | SGD |
| | # hidden layers | 1 |
| Regression on $q^*$ with 1 layer MLP | Learning rate | 0.1 |
| | Batch size | 1 |
| | Optimiser | SGD |
| | # hidden layers | 1 |
| Q-learning with 3 layer MLP | Learning rate | 0.1 |
| | Batch size | 1 |
| | Optimiser | SGD |
| | # hidden layers | 3 |
| DQN-like with RMSProp | Learning rate | 0.001 |
| | Batch size | 32 |
| | Optimiser | RMSProp |
| | Optimiser $\varepsilon$ | $10^{-5}$ |
| | Replay capacity | 1 000 |
| | # hidden layers | 3 |
| DQN-like with SGD | Learning rate | 0.01 |
| | Batch size | 32 |
| | Optimiser | SGD |
| | Replay capacity | 1 000 |
| | # hidden layers | 3 |
| DQN-like with Adam | Learning rate | 0.001 |
| | Batch size | 32 |
| | Optimiser | Adam |
| | Optimiser $\varepsilon$ | $10^{-8}$ |
| | Replay capacity | 1 000 |
| | # hidden layers | 3 |
| (Common hyper-parameters) | Exploration $\epsilon$ | 0.1 |
| | # units per hidden layer | 25 |

Table 3: CATCH case study variant settings. These are the relevant settings for the variants used to generate Figure 4.

convergence is defined as when the greedy policy achieves the maximum score for 100 evaluation episodes. Convergence is periodically tested every 100 training episodes.

## B.4   Dynamic programming

To measure policy change of dynamic programming in a tabular MDP, we exploit the knowledge of the exact transition dynamics, encoded via a matrix $\mathbf{T}$ to compute value or policy iteration updates that do not involve sampling or interactions. Values are initialised at $0$, and for the purposes of measuring policy change, all $\arg\max$ actions whose Q-values are exactly tied also share equal probability mass. As example domain we use a $16 \times 16$ Gridworld with 4-room structure, initial state in one corner, goal state in opposing corner and $\gamma = 0.97$. Figure 13 (left) shows the amounts of policy change accumulated in such a process.

## B.5   MNIST experiments

For a simple initial supervised learning experiment, we used an off-the-shelf neural network training setup on MNIST. Thus we used a 3-layer MLP with 300 and 100 hidden units, ReLU non-linearities, a softmax output, cross-entropy loss and the Adam optimiser [23]. Policy change is measured on the softmax probability outputs of the classification network, with equal weight on all samples of the test

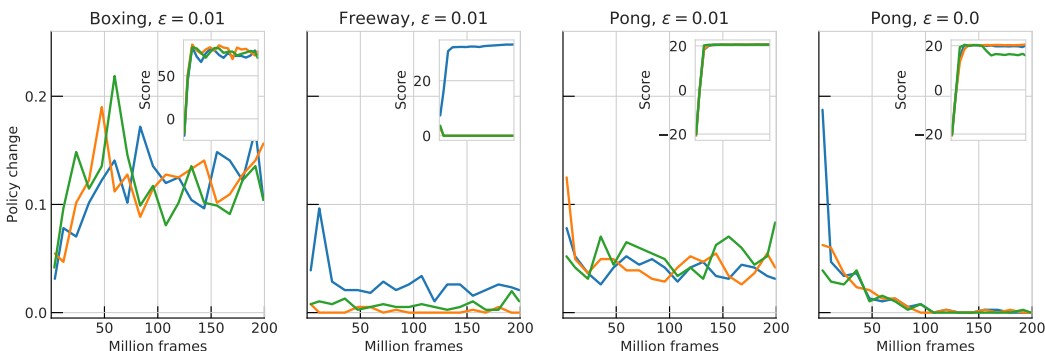

Figure 14: Policy change on plateaus. We observe a high amount of policy change (per single update, i.e., $\overline{W}(\pi_t, \pi_{t+1})$) even in periods where overall policy performance is flat (see performance curves on inset plots). Each curve corresponds to a single run (seed) and is smoothed over 10M frames. An interesting effect to highlight is in FREEWAY, where one seed (blue) converges to high performance and the other two seeds collapse to zero performance, and the "broken" runs also have much lower churn. The right-most figure shows (on PONG) that converged performance, together with $\epsilon = 0$ leads to policy change that eventually *does* seem to decay.

set. It is accumulated across all $P$ gradient updates. Our experiments are stopped when reaching $2\%$ training error, which happens after $P \approx 1\,000 - 10\,000$ updates. Figure 13 (right) shows the results.

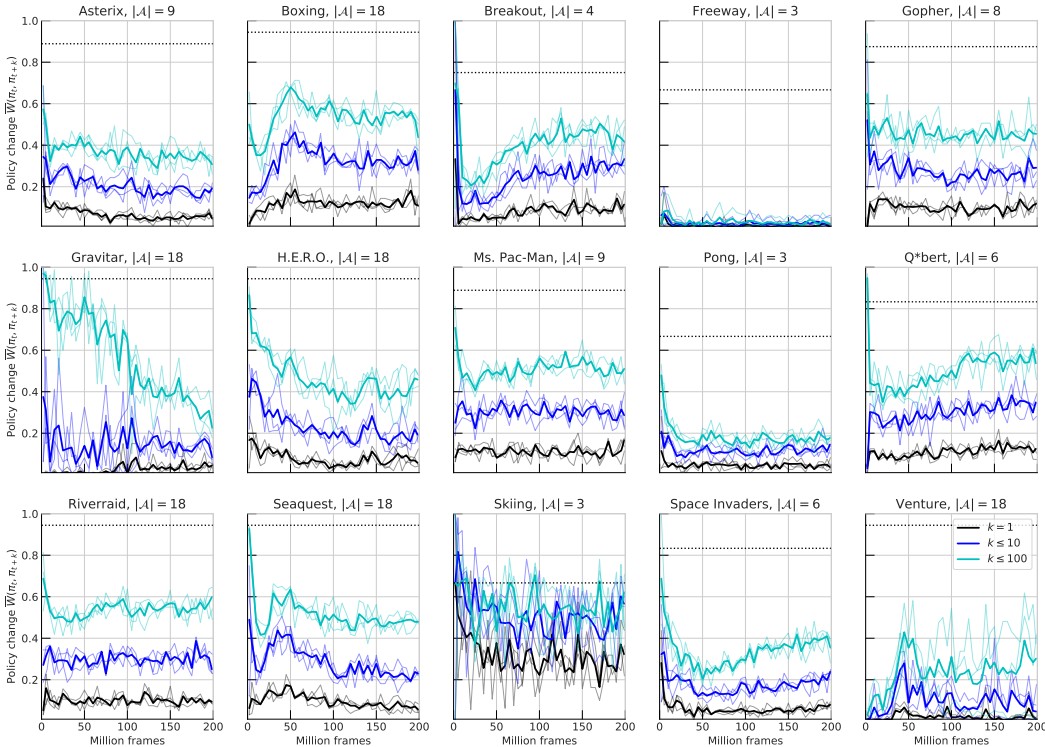

Figure 15: Average policy change $\overline{W}$ as a function of training stage in DoubleDQN, across 15 Atari games. Often, but not always, policy change is larger in early learning. Different colours show different interval sizes $k$ across which $\overline{W}(\pi_t, \pi_{t+k})$ is measured. In some scenarios these show a more cumulative effect (e.g., GRAVITAR), in others the change between the very next policy ($k = 1$) is almost as large to the change after $k = 100$ updates, as in SKIING. Dotted lines indicate what "maximal" policy change would look like for a given action space size $|\mathcal{A}|$, i.e., if $\arg\max$ actions were completely random. Thin lines are individual seeds (3), thick lines their average. See Figure 14 for a detailed look at the low levels of policy change after convergence as in PONG or FREEWAY.

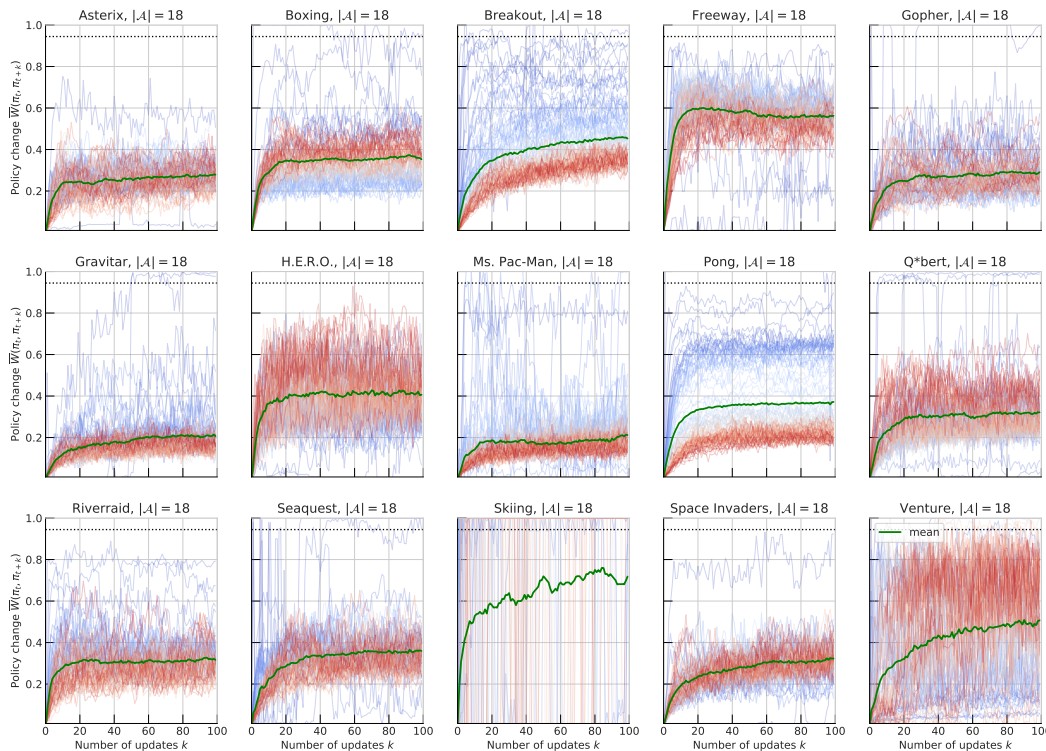

Figure 16: Average policy change $\overline{W}(\pi_t, \pi_{t+k})$ as a function of the number of in-between updates $k$. In contrast to Figure 15, these results are from the R2D2 agent (always using the full action set of $|\mathcal{A}| = 18$). Thick green lines show the average across training, while thin lines show snapshots from different points in training, with cooler and warmer colors denoting early and late stages of training respectively. We can see how policy change quickly rises and then saturates, generally between 20% and 60%. This means that, compared to $\pi_t$, the policy $\pi_{t+100}$ generally does not differ much more than $\pi_{t+20}$. This is consistent with the hypothesis that policy churn only affects a subset of states. An outlier here are the SKIING results, where the observed fraction of $\arg\max$ switches (in a minibatch of $32 \times 80$ states) is always either 0 or 1: this seems to indicate that the Q-values have essentially no state-dependence (note that performance also does not take off in this game, see Figure 24).

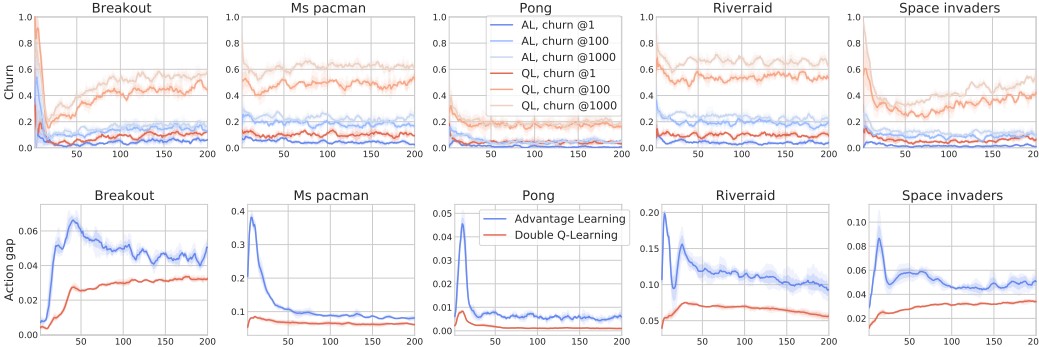

Figure 17: Double Q-Learning ("QL", red) versus action-gap-increasing Advantage Learning ("AL", blue), on a set of 5 games. **Top:** policy change, where "@$k$" denotes the interval in $\overline{W}(\pi_t, \pi_{t+k})$. **Bottom:** corresponding action gaps. This provides time-series detail to Figure 3 (right).

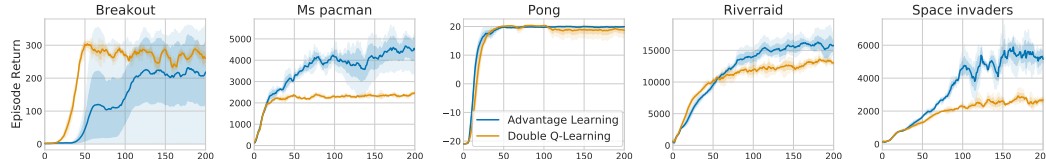

Figure 18: Performance results of Double Q-Learning and Advantage Learning, as in Figure 17, but in the $\epsilon = 0$ setting. Despite reduced churn, Advantage Learning is the higher-performing algorithm, indicating that not the full amount of DoubleDQN's observed policy change is needed for performance, even in the absence of other forms of exploration. This matches the insights in Figure 21.

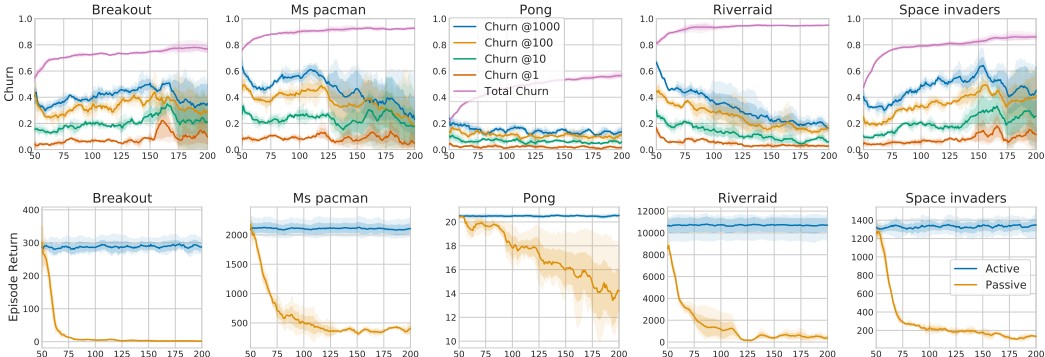

Figure 19: Stationary data in the "forked tandem" setting: after 50M frames (start of $x$-axis), a passive learner is forked off, which means that it does not influence behaviour anymore (and cannot self-correct). It receives a data stream from a fixed, frozen policy network. **Top:** active vs. passive performance. **Bottom:** policy change. the purple curve ("total churn") denotes the difference between the active (frozen) policy and the current policy of the passive (but learning) network.

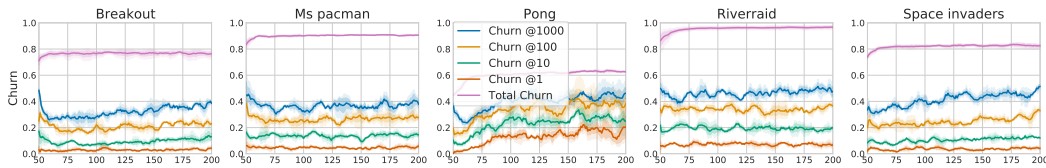

Figure 20: Stationary data *and targets*. Setup as in Figure 19, but instead of Q-learning bootstrap targets, stationary regression targets are constructed from Monte-Carlo returns.

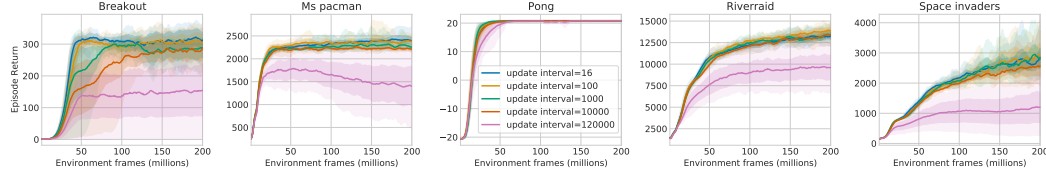

Figure 21: Ablation experiment with a separate copy of the Q-network used exclusively for acting; this network is a periodic copy of the online (learning) network, just like the target network, but updated at a different frequency. "Interval $= 16$" corresponds to the DoubleDQN baseline, while "Interval $= 120\,000$" corresponds to the "act with target network" of Section 2 and Figure 2 (denoted "no churn" there). We find again (cf. Figure 18) that the full empirical magnitude of policy change in DoubleDQN is not needed for exploration: reducing the number of different greedy policies used for acting by a factor $100 - 1\,000$ still retains a very similar exploration effect.

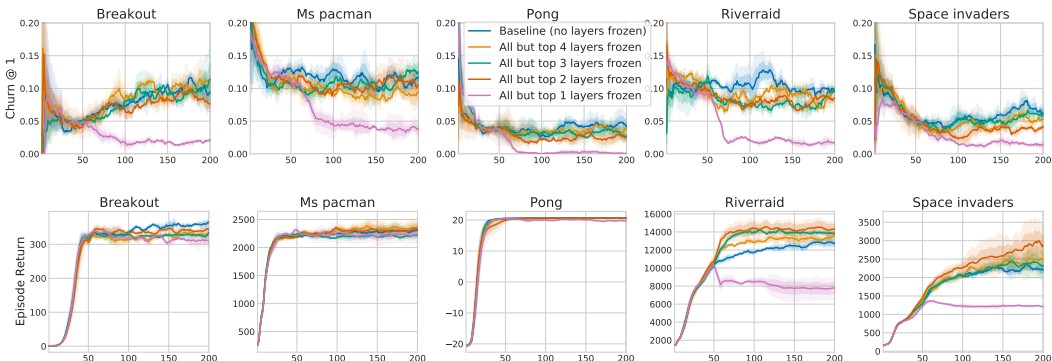

Figure 22: Ablation experiment that relates the depth of the neural network being trained to the amount of policy change. After 50M frames of regular training, all but a few top layers of DoubleDQN's neural network are frozen, and the remainder of training can only change weights in the last $1 - 4$ layers. We find a correlation between churn and trainable capacity, but the most significant step-change occurs between one or more trainable layers, i.e., between linear FA (on top of frozen features) and deep learning.

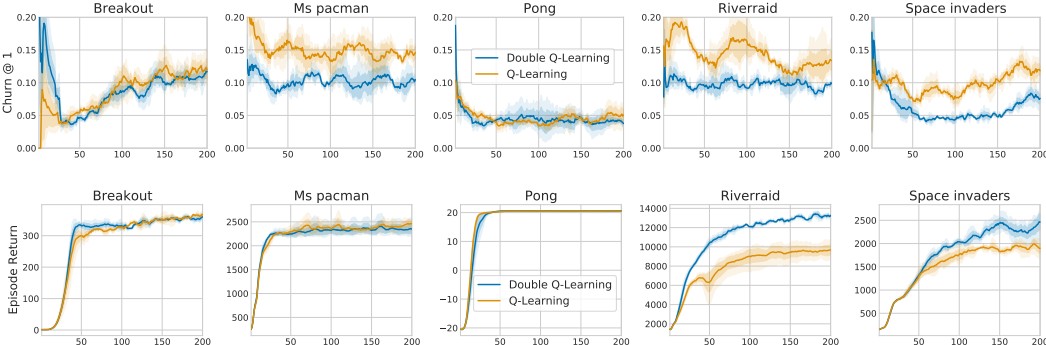

Figure 23: DQN versus DoubleDQN. Overall, DoubleDQN has somewhat better performance, while the level of policy change is a bit lower but not drastically different; in fact, the variation across games or across learning stages tends to be larger than the difference between algorithms.

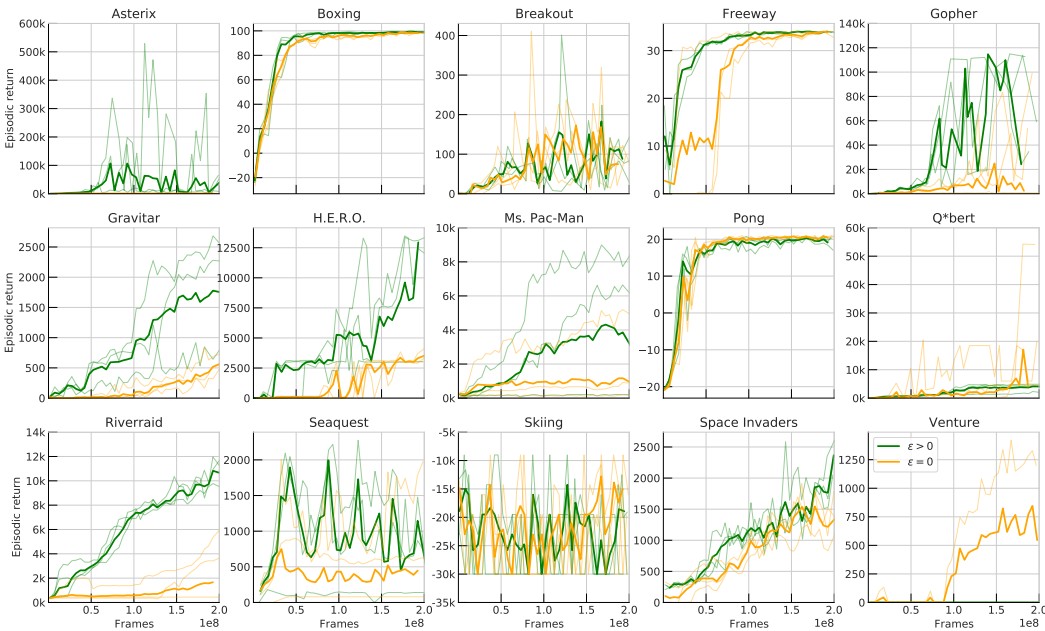

Figure 24: R2D2 performance curves. The setting is the same as in Figure 2, namely 200M frames, 15 games, 3 seeds each (thin lines), but the agent architecture is very different (see Table 1). In comparison, the R2D2 agent is less robust to $\epsilon = 0$; despite high policy change, exploration appears to suffer in half of the games. We assume this difference is mainly due to two aspects: first, DoubleDQN has a high amount of random exploration in early learning (it takes 4M frames until $\epsilon$ has decayed to 0). Second, DoubleDQN traverses many more distinct policy networks over the course of its lifetime ($\approx 10^7$), compared to R2D2 ($\approx 10^4$), due to the latter's much larger batch size, greater parallelism, and smaller replay ratio. Note also that the maximal "policy age" (in gradient updates) and as a consequence policy diversity representend in the replay buffer data is very different in R2D2 and DQN. Because of the data generation parallelism (and the near-deterministic dynamics of the Atari environment), diversity of replay data in R2D2 may be driven more by $\epsilon$-exploration than in DQN. The case $\epsilon = 0$ may therefore result in a very narrow data distribution and potentially collapse of the neural network representation in R2D2.

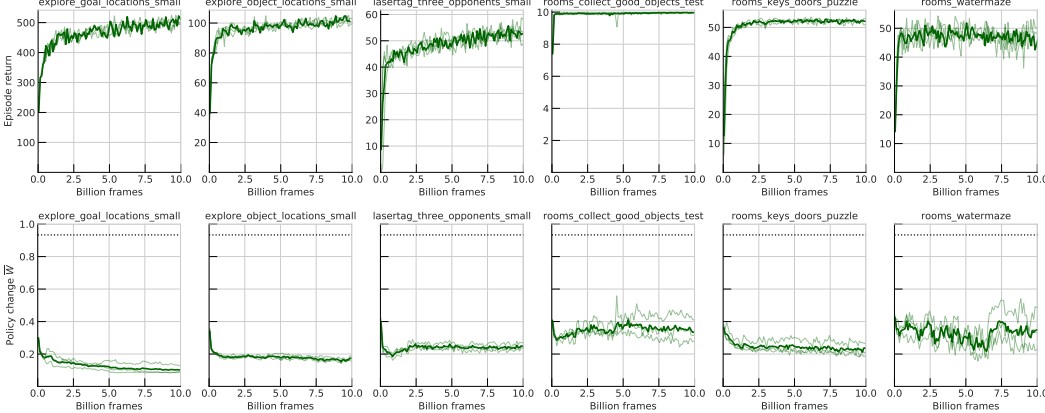

Figure 25: Experiments using R2D2 [21] on 6 levels of the DM-Lab [1] benchmark suite of 3D environments (3 seeds of 10B frames each), with $|\mathcal{A}| = 15$. The average observed policy change ($\approx 20\%$, see bottom plots) is overall in line with the Atari results, but somewhat higher, possibly because of the different action space, where most actions can be easily undone at the next step.

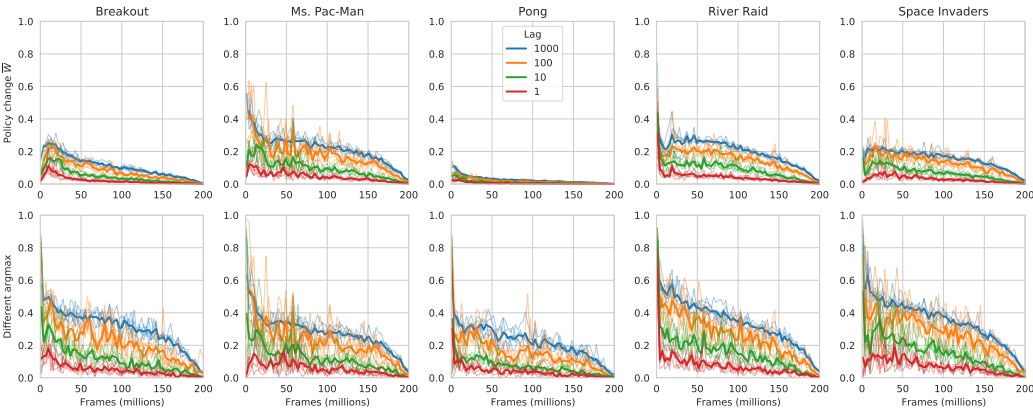

Figure 26: Preliminary experiments with an **actor-critic** agent on a subset of Atari games (5 seeds each), with minimal action spaces per game (as in the DDQN setup, see Figure 15). The agent is an implementation of IMPALA [14] in the Sebulba framework [19]. The policy change is comparable to other agents on Atari, showing policy churn is present in actor-critic agents, not just value-based agents. Note policy change reduces to zero as training progresses because the learning rate is linearly annealed to zero. **Top row**: Total variation policy change $\overline{W}$, as defined in Eqs. 1 and 2; as these are soft policies, the change is expected to be smaller than it would be for switches between greedy policies. **Bottom row**: Shows the average $\arg\max$ switches on the same experiment. Different colours show different intervals $k$ across which $\overline{W}(\pi_t, \pi_{t+k})$ is measured (as in Figure 15).