# OpenReview forum: "The Phenomenon of Policy Churn"
_NeurIPS.cc/2022/Conference — NeurIPS 2022 Accept_

### Official Review · Reviewer_Q1cd · 2022-07-06

**Rating:** 8
**Confidence:** 4
**Soundness:** 4 excellent
**Presentation:** 4 excellent
**Contribution:** 4 excellent

**Summary:**

This paper identifies and studies a phenomena they name policy churn; the fact that during the learning process in value-based RL algorithms, the greedy policy changes at a very rapid rate (in a single update the greedy policy changes in 10% of all states). They observe and justify empirically that this phenomena is a key driver of exploration in Atari games. They identify this phenomena as happening both in DoubleDQN and R2D2, which are fairly different value-based algorithms, demonstrating that the phenomena is fairly general. The investigate several possible causes of the phenomena, and while they don't fully explain it, they propose a possible hypothesis for the primary causes of policy churn (non-linear global function approximation combined with a highly noisy learning process). They provide an extended discussion on the implications and future directions of research from this phenomena.

**Questions:**

I don't have any low-level questions. My main high-level questions are mostly posed in the strengths and weaknesses section, in terms of asking for additional experiments.

**Limitations:**

While there is some implicit discussion of limitations throughout the paper, I think it would be beneficial to explicitly state that it's not yet clear that this phenomena is more general than off-policy value-based methods on Atari.

**Strengths And Weaknesses:**

Overall I like the paper. I think exploratory and explanatory work like this is often undervalued, but is crucial to improving our understanding of how RL algorithms work. The paper is well-written and easy to follow, and clearly demonstrates an interesting phenomena, while backing it up with a wide variety of empirical experiments and investigations. The research claims made are sound, and appropriately cautious where there is not compelling evidence. For all these reasons, I'm in favour of accepting the paper.

The main weakness lies in the fact that the phenomena isn't entirely explained. I think this is too-high a bar to require for the paper's acceptance, but it would be great to have a more complete explanation of the phenomena.

A second weakness that I think is possibly more easily addressable the lack of generality in the environments on which the phenomena is investigated. While Atari is a staple of the RL literature, I'm cautious of making general claims about RL learning from performing experiments on a single set of environments, which while diverse in some ways are very non-diverse in others. While I'd expect this same phenomena to hold for value-based methods applied in other environments, I'd prefer to have the prediction validated, and it seems like a relatively easy set of experiments to perform.

I'd also be interested in seeing investigation of policy churn in on-policy methods (such as PPO), or in the policies of actor-critic methods for continuous control such soft actor critic or TD3. While I think the contribution is still valuable even if it's limited to off-policy value-based methods similar to DQN, showing that the phenomena is more general than that would be beneficial.

I'm currently recommending accept, and would raise my rating to a strong accept if experiments were performed on other environments, either with similar or different methods, regardless of the outcomes of those experiments. I'd also raise my score if the mechanistic hypothesis in 3.4 was operationalised technically, and then investigated, although I'm not expecting that as it requires more work that can reasonably be expected in a rebuttal period.

EDIT: I thank the authors for their response, and the addition of new experiments demonstrating the broad appearance of the policy churn phenomena. I raise my score to an 8 in light of this.

---

> ### Author Response · Authors · 2022-08-02
> **Added experiments on different environments and actor-critic agents show broad prevalence of the phenomenon.**
>
> >  [...] the environments on which the phenomena is investigated. While Atari is a staple of the RL literature, I'm cautious of making general claims about RL learning from performing experiments on a single set of environments, which while diverse in some ways are very non-diverse in others. While I'd expect this same phenomena to hold for value-based methods applied in other environments, I'd prefer to have the prediction validated, and it seems like a relatively easy set of experiments to perform.
>
> The updated version of the paper now also includes a number of **DMLab** experiments, a 3D domain with quite distinct properties from Atari, but comparably high levels of policy churn (Fig 25).
>
> > I'd also be interested in seeing investigation of policy churn in on-policy methods (such as PPO), or in the policies of actor-critic methods for continuous control such soft actor critic or TD3. While I think the contribution is still valuable even if it's limited to off-policy value-based methods similar to DQN, showing that the phenomena is more general than that would be beneficial.
>
> The updated version of the paper now also includes a number of **actor-critic** experiments (near on-policy, using an IMPALA variant), again with comparable levels of policy churn (Fig 26).
>
> > I think it would be beneficial to explicitly state that it's not yet clear that this phenomena is more general than off-policy value-based methods on Atari.
>
> We thank the reviewer for this important remark. While our second-to-last paragraph (“Churn beyond value-based RL”) was aiming to spell out this limitation, with the added actor-critic and DMLab results, we believe we can now claim that the phenomenon is more broadly prevalent in deep RL.

---

> > ### Comment · Reviewer_Q1cd · 2022-08-05
> > **Response**
> >
> > I thank the authors for their response, and the addition of new experiments demonstrating the broad appearance of the policy churn phenomena. I raise my score to an 8 in light of this.

---

### Official Review · Reviewer_ZHm5 · 2022-07-12

**Rating:** 6
**Confidence:** 4
**Soundness:** 3 good
**Presentation:** 3 good
**Contribution:** 3 good

**Summary:**

This paper firstly identifies the policy churn phenomenon, which is not noticed by RL practitioners. The paper demonstrates that this phenomenon is related to deep learning. Furthermore, they found that this phenomenon is a benefit for exploration.

**Questions:**


Questions:
1. In Fig. 2, what is a policy change for only churn? How does it compare the policy change of "both $\epsilon$ and churn" and "only $\epsilon>0$"

2. Policy churn is not always a good thing. For example, there should be less policy churn when converging. Could the author discuss more on how to control policy churn when the policy converges? Could that improve the final performance if the policy churn is controlled?

3. In line 323,  I don't know why the paper states that policy churn is complementary to simpler noise-based exploration. At least in Fig 2, combining noise-based and churn didn't perform best on all tasks (either does policy churn).

4. in my view, policy churn is still different from noise-based exploration, as noise-based exploration imposes the largest probability of the most promising action. Could the policy churn attain these effects, i.e., controllable exploration? Did the author do experiments on that?



Minor Problems:

$\bar{W}^+$ is a variable related to $P$. Would that be better to use a subscript with P?


**Limitations:**



**Strengths And Weaknesses:**



Generally, I like the work of this paper. Policy churn is not noticed by RL practitioners previously. This paper did good work on this interesting phenomenon. It provided abundant experimental analysis.
The results make sense to me. It's expected in a dynamic programming setting that policy churn is not significant. In contrast, the noise in updating can result in this phenomenon.

The main effect that this paper study is to relate policy churn with exploration, which is interesting. However, there are some flaws in the statements (see questions below). Furthermore, I expect the author to discuss more whether policy churn takes effect on other elements of RL, e.g., policy improvement, and the quality of resulted policy.

---

> ### Author Response · Authors · 2022-08-02
> **Clarifications about complementarity, usefulness (or not) and controllability of churn.**
>
> > 1. In Fig. 2, what is a policy change for only churn? How does it compare the policy change of "both ϵ and churn" and "only ϵ>0"
>
> Policy churn is generally similar across these two scenarios (with and without ε). Fig 5 (black) shows the detailed plots for the “both ϵ and churn” variant, and Fig 7 (two rightmost subplots about Pong) shows one direct comparison between them, in the specific scenario after convergence to the optimal performance: here the ε=0 churn is significantly smaller (but outside of convergence, which is the more typical scenario, the two settings have similar levels of churn).
>
> > 2. Policy churn is not always a good thing. For example, there should be less policy churn when converging. Could the author discuss more on how to control policy churn when the policy converges? Could that improve the final performance if the policy churn is controlled?
>
> Indeed, policy churn is not always a good thing. In some scenarios, churn is decreasing after convergence (see Fig 7, right-most subplot with ε=0). We also added the new Fig 24, which shows how late in training on catch, policy change starts concentrating in the null space (where action gaps are zero) and critical actions are perturbed less and less often. Furthermore, the added Fig 26 shows the effect of learning rate annealing: as expected, this reduces policy churn.
>
> > 3. In line 323, I don't know why the paper states that policy churn is complementary to simpler noise-based exploration. At least in Fig 2, combining noise-based and churn didn't perform best on all tasks (either does policy churn).
> > 4. in my view, policy churn is still different from noise-based exploration, as noise-based exploration imposes the largest probability of the most promising action. Could the policy churn attain these effects, i.e., controllable exploration? Did the author do experiments on that?
>
>
> We did not claim that churn and noise **are** complementary, but that they **could** be (we have adjusted phrasing to make this clearer). So far we do not have strong evidence for or against complementarity. What we do know is that the two kinds of exploration have different properties (see also the new Fig 24). In our view, the most satisfying way to answer this question would be designing a fully controlled, explicit algorithmic variant of policy churn (as alluded to in our last paragraph, and in contrast to churn as a side-effect of learning) and then tune such an agent with or without additional noise-based exploration: it could be that together churn and noise is an overall better exploration strategy than either in isolation. We have done a few (simple) preliminary experiments to obtain such a fully controlled variant of churn, but without success so far.

---

### Official Review · Reviewer_6hS6 · 2022-07-13

**Rating:** 5
**Confidence:** 3
**Soundness:** 3 good
**Presentation:** 3 good
**Contribution:** 3 good

**Summary:**

This work identified and studied the phenomenon of policy churn, which refers to the rapid change of the greedy policy in value-based reinforcement learning.

The authors first conducted experiments on several Atari games with two methods (DoubleDQN and R2D2), and the results in Figure 1 showed that each one update would cause about $10$% greedy policy changes of all states during training, while $5$% greedy policy changes when the performance of the policy no longer changes.

The authors then showed that the policy churn has some exploration effect. As shown in Figure 2, acting with the policies produced by a learning processing with high churn would make additional eps-greedy exploration kind of not necessary. And removing the churn would reduce performance.

Section 3 used ablations to eliminate several potential reasons for policy churn to happen, and made some hypotheses that policy churn is a result of non linear function approximation and stochastic updates.

**Questions:**

1. It is claimed that "removing the churn would reduce performance". However, as shown in Figure 2, it seems that for many games the blue curves are comparable to green and gold ones? How is this concluded from this figure?

2. The way the authors did to "removing the churn" is "acting with the target network". As noted this also changed the learning dynamics somehow rather than just "removing the churn". This seems to introduce a compounding effect: by "acting with the target network", the samples being collected are different, and thus it changes the policies after performing updates using the collected samples.

In a word, "acting with the target network" introduced other changes rather than "removing the churn", and it seems "shifting the x-axis of the blue curve by 120000 frames to the left" is not convincing to me to correct it. Therefore, this part of argument for "policy churn has exploration effect" is not convincing to me.

Please correct me if I misunderstood something or please clarify this matter.

------update------
Thank you for the response. I would increase my score since I agree with the authors and other reviewers that discovering the policy churn phenomenon would be valuable and could inspire future study.

**Limitations:**

The experiments are conducted on publicly available benchmarks which have been widely used for years. I did not see negative societal impact of this work.

**Strengths And Weaknesses:**

Strengths:

1. Policy churn of value-based RL methods is an unnoticed phenomenon, and it is to me an interesting and relevant result.

2. The experiments and ablations are substantive, which I appreciate.

3. The presentation is clear and easy to follow.

Weaknesses:

1. It remains still unclear as noted how the policy churn phenomenon can be useful for designing better algorithms.

2. It is still largely unclear where the concrete root cause of the policy churn is.

3. The argument that policy churn is a good way of exploration is not very convincing.

---

> ### Author Response · Authors · 2022-08-02
> **Response to reviewer questions.**
>
> > It is still largely unclear where the concrete root cause of the policy churn is.
>
> Indeed, there remain open questions about the phenomenon, but we would like to stress the many contributions that this paper makes in getting to the bottom of it. We conducted dozens of ablations and detailed analyses with multiple agents, optimisers, architectures and experimental setups on multiple domains, and have been able to eliminate a large number of potential causes and mechanisms that way.
>
> > The argument that policy churn is a good way of exploration is not very convincing.
>
> We agree with the reviewer that policy churn may not be a good way to explore, and did not intend to imply that in our work (the text does not describe churn as a **good** way to explore anywhere).  The message of the paper here is that churn does contribute to an agent’s exploration, in a way that is different from ε-greedy. Moreover, as indicated in Section 4.4, we believe that churn as a de-facto contributor to exploration may have (unintentionally) found its way into existing RL agents, with other algorithmic components over time implicitly “co-adapting” to this particular, perhaps suboptimal, mode of exploration.
>
> > 1. It is claimed that "removing the churn would reduce performance". However, as shown in Figure 2, it seems that for many games the blue curves are comparable to green and gold ones? How is this concluded from this figure?
>
> The comparison here is between green (ε with churn) and blue (ε without churn), and Fig 2 shows on 7 games that there is a small but meaningful drop when removing churn. Following your suggestion, we have updated this section with the relevant line now reading “**sometimes** reduces performance”. Note that the paper’s core messages do not hinge on this observation; if green were equivalent to blue, the rest would still stand.
>
> > 2. The way the authors did to "removing the churn" is "acting with the target network". As noted this also changed the learning dynamics somehow rather than just "removing the churn". This seems to introduce a compounding effect: by "acting with the target network", the samples being collected are different, and thus it changes the policies after performing updates using the collected samples.
>
> > In a word, "acting with the target network" introduced other changes rather than "removing the churn", and it seems "shifting the x-axis of the blue curve by 120000 frames to the left" is not convincing to me to correct it. Therefore, this part of argument for "policy churn has exploration effect" is not convincing to me.
>
> We thank the reviewer for this important observation: Indeed, as pointed out, in RL, exploration and learning are unavoidably entangled, so changing the acting behaviour always influences the whole system, with compounding effects on future policies and future learning. In particular, the footnote about shifting the x-axis is only making a statement about one specific kind of latency that can be compensated that way, not that this trick allows for a perfect disentangling of the churn effect. We have clarified the corresponding footnote, hopefully removing the ambiguity.

---

### Author Response · Authors · 2022-08-02
**Response to all reviewers, summary of revision, new experiments.**

We thank all three reviewers for the thoughtful and constructive feedback, and are happy to be able to address all of the concerns in detail below.

One concern that shows up in two reviews (6hS6 and Q1cd) is the lack of definite explanation for the phenomenon of policy churn. Although we acknowledge that the phenomenon is not fully explained yet, we argue that pointing out its existence, which had passed unnoticed so far, is an important contribution in its own right. We also note that we do present a testable hypothesis as to why policy churn happens, and such hypothesis can serve as a starting point for future investigations in this area.

Another general point we would like to emphasise is that the effect of policy churn on exploration is one of possibly many manifestations of the phenomenon in practice. The extent to which such an effect is beneficial is less important than its very existence, which, to the best of our knowledge, was also unknown to the community. By calling attention to this effect, we invite future investigations not only on exploration itself, but also on other potential side effects of policy churn that have passed unnoticed to us.

Finally, we would like to give a high level overview of the changes we have made to the paper, which now presents a much broader view of policy churn. The updated version of the paper now includes results on policy churn on different environments (**DMLab**, Fig 25), and different algorithm types (**actor-critic**, Fig 26), as well as refined claims and additional analysis (eg Fig 24). The new results confirm our original claims in a broader set of scenarios: policy churn seems to show up wherever we look in deep RL.

---

### Meta-Review · Area_Chair_phLH · 2022-08-27

**Recommendation:** Accept
**Confidence:** Certain

**Metareview:**

The paper identifies a hitherto unknown phenomenon in value-based reinforcement learning called policy churn. All the reviewers agree that this is a very interesting phenomenon, and that the paper studies the phenomenon comprehensively. All reviewers also agreed that this paper is likely to inspire many follow-up works. Understanding the causes and harnessing policy churn can potentially significantly impact the deep RL state-of-the-art.

**Award:**

No

---

### Decision · Program_Chairs · 2022-09-14

Accept